# Uniform Convergence with Square-Root Lipschitz Loss

**Lijia Zhou**
University of Chicago
zlj@uchicago.edu

**Zhen Dai**
University of Chicago
zhen9@uchicago.edu

**Frederic Koehler**
Stanford University
fkoehler@stanford.edu

**Nathan Srebro**
Toyota Technological Institute at Chicago
nati@ttic.edu

Collaboration on the Theoretical Foundations of Deep Learning (deepfoundations.ai)

## Abstract

We establish generic uniform convergence guarantees for Gaussian data in terms of the Rademacher complexity of the hypothesis class and the Lipschitz constant of the square root of the scalar loss function. We show how these guarantees substantially generalize previous results based on smoothness (Lipschitz constant of the derivative), and allow us to handle the broader class of square-root-Lipschitz losses, which includes also non-smooth loss functions appropriate for studying phase retrieval and ReLU regression, as well as rederive and better understand "optimistic rate" and interpolation learning guarantees.

## 1 Introduction

The phenomenon of "interpolation learning", where a learning algorithm perfectly interpolates noisy training labels while still generalizing well to unseen data, has attracted significant interest in recent years. As Shamir (2022) pointed out, one of the difficulties in understanding interpolation learning is that we need to determine which loss function we should use to analyze the test error. In linear regression, interpolating the square loss is equivalent to interpolating many other losses (such as the absolute loss) on the training set. Similarly, in the context of linear classification, many works (Soudry et al. 2018; Ji and Telgarsky 2019; Muthukumar et al. 2021) have shown that optimizing the logistic, exponential, or even square loss would all lead to the maximum margin solution, which interpolates the hinge loss. Even though there are many possible loss functions to choose from to analyze the population error, consistency is often possible with respect to only one loss function because different loss functions generally have different population error minimizers.

In linear regression, it has been shown that minimal norm interpolant can be consistent with respect to the square loss (Bartlett et al. 2020; Muthukumar et al. 2020; Negrea et al. 2020; Koehler et al. 2021), while the appropriate loss for benign overfitting in linear classification is the squared hinge loss (Shamir 2022; Zhou et al. 2022). In this line of work, uniform convergence has emerged as a general and useful technique to understand interpolation learning (Koehler et al. 2021; Zhou et al. 2022). Though the Lipschitz contraction technique has been very useful to establish uniform convergence in classical learning theory, the appropriate loss functions to study interpolation learning (such as the square loss and the squared hinge loss) are usually not Lipschitz. In fact, many papers (Nagarajan and Kolter 2019; Zhou et al. 2020; Negrea et al. 2020) have shown that the traditional notion of uniform convergence implied by Lipschitz contraction fails in the setting of interpolation learning. Instead, we need to find other properties of the loss function to establish a different type of uniform convergence.

37th Conference on Neural Information Processing Systems (NeurIPS 2023).

Zhou et al. (2022) recently showed that the Moreau envelope of the square loss or the squared hinge loss is proportional to itself and this property is sufficient to establish a type of uniform convergence guarantee known as "optimistic rate" (Panchenko 2002; Srebro et al. 2010). Roughly speaking, if this condition holds, then the difference between the *square roots* of the population error and the training error can be bounded by the Rademacher complexity of the hypothesis class. Specializing to predictors with zero training error, the test error can then be controlled by the square of the Rademacher complexity, which exactly matches the Bayes error for the minimal norm interpolant when it is consistent (Koehler et al. 2021). However, the Moreau envelope of a loss function is not always easy to solve in closed-form, and it is unclear whether this argument can be applied to problems beyond linear regression and max-margin classification. Moreover, considering the difference of square roots seem like a mysterious choice and does not have a good intuitive explanation.

In this paper, by showing that optimistic rate holds for any *square-root Lipschitz* loss, we argue that the class of square-root Lipschitz losses is the more natural class of loss functions to consider. In fact, any loss whose Moreau envelope is proportional to itself is square-root Lipschitz. Our result also provides an intuitive explanation for optimistic rate: when the loss is Lipschitz, the difference between the test and training error can be bounded by Rademacher complexity; therefore, when the loss is *square-root Lipschitz*, the difference between the *square roots* of the test and training error can be bounded by Rademacher complexity. In addition to avoiding any hidden constant and logarithmic factor, our uniform convergence guarantee substantially generalize previous results based on smoothness (Lipschitz constant of the derivative, such as Srebro et al. 2010). The generality of our results allows us to handle non-differentiable losses. In the problems of phase retrieval and ReLU regression, we identify the consistent loss through a norm calculation, and we apply our theory to prove novel consistency results. In the context of noisy matrix sensing, we also establish benign overfitting for the minimal nuclear norm solution. Finally, we show that our results extend to fully optimized neural networks with weight sharing in the first layer.

## 2   Related Work

**Optimistic rates.**   In the context of generalization theory, our results are related to a phenomena known as "optimistic rates" — generalization bounds which give strengthened guarantees for predictors with smaller training error. Such bounds can be contrasted with "pessimistic" bounds which do not attempt to adapt to the training error of the predictor. See for example Vapnik 1982; Panchenko 2003; Panchenko 2002. The work of Srebro et al. 2010 showed that optimistic rates control the generalization error of function classes learned with smooth losses, and recently the works Zhou et al. 2021; Koehler et al. 2021; Zhou et al. 2020 showed that a much sharper version of optimism can naturally explain the phenomena of benign overfitting in Gaussian linear models. (These works are in turn connected with a celebrated line of work in proportional asymptotics for M-estimation such as Stojnic 2013, see references within.) In the present work, we show that the natural setting for optimistic rates is actually square-root-Lipschitz losses and this allows us to capture new applications such as phase retrieval where the loss is not smooth.

**Phase retrieval, ReLU Regression, Matrix sensing.**   Recent works (Maillard et al. 2020; Barbier et al. 2019; Mondelli and Montanari 2018; Luo et al. 2019) analyzed the statistical limits of phase retrieval in a high-dimensional limit for certain types of sensing designs (e.g. i.i.d. Real or Complex Gaussian), as well as the performance of Approximate Message Passing in this problem (which is used to predict a computational-statistical gap). In the noiseless case, the works Andoni et al. 2017; Song et al. 2021 gave better computationally efficient estimators for phase retrieval based upon the LLL algorithm. Our generalization bound can naturally be applied to any estimator for phase retrieval, including these methods or other ones, such as those based on non-convex optimization (Sun et al. 2018; Netrapalli et al. 2013). For the same reason, they can also naturally be applied in the context of sparse phase retrieval (see e.g. Li and Voroninski 2013; Candes et al. 2015).

Similarly, there have been many works studying the computational tractability of ReLU regression. See for example the work of Auer et al. 1995 where the "matching loss" technique was developed for learning well-specified GLMs including ReLUs. Under misspecification, learning ReLUs can be hard and approximate results have been established, see e.g. Goel et al. 2017; Diakonikolas et al. 2020. In particular, learning a ReLU agnostically in squared loss, even over Gaussian marginals, is known to be hard under standard average-case complexity assumptions (Goel et al. 2019). Again, our

generalization bound has the merit that it can be applied to any estimator. For matrix sensing, we consider nuclear norm minimization (as in e.g. Recht et al. 2010) which is a convex program, so it can always be computed in polynomial time.

**Weight-tied neural networks.** The work of Bietti et al. 2022 studies a similar two-layer neural network model with weight tying. Their focus is primarily on understanding the non-convex optimization of this model via gradient-based methods. Again, our generalization bound can be straightforwardly combined with the output of their algorithm. The use of norm-based bounds as in our result is common in the generalization theory for neural networks, see e.g. Bartlett (1996) and Anthony, Bartlett, et al. (1999). Compared to existing generalization bounds, the key advantage of our bound is its quantitative sharpness (e.g. small constants and no extra logarithmic factors).

## 3   Problem Formulation

In most of this paper, we consider the setting of generalized linear models. Given any loss function $f : \mathbb{R} \times \mathcal{Y} \to \mathbb{R}_{\geq 0}$ and independent sample pairs $(x_i, y_i)$ from some data distribution $\mathcal{D}$ over $\mathbb{R}^d \times \mathcal{Y}$, we can learn a linear model $(\hat{w}, \hat{b})$ by minimizing the empirical loss $\hat{L}_f$ with the goal of achieving small population loss $L_f$:

$$\hat{L}_f(w, b) = \frac{1}{n} \sum_{i=1}^{n} f(\langle w, x_i \rangle + b, y_i), \quad L_f(w, b) = \mathbb{E}_{(x,y) \sim \mathcal{D}}[f(\langle w, x \rangle + b, y)]. \quad (1)$$

In the above, $\mathcal{Y}$ is an abstract label space. For example, $\mathcal{Y} = \mathbb{R}$ for linear regression and $\mathcal{Y} = \mathbb{R}_{\geq 0}$ for phase retrieval (section 5.1) and ReLU regression (section 5.2). If we view $x_i$ as vectorization of the sensing matrices, then our setting (1) also includes the problem of matrix sensing (section 5.3).

For technical reasons, we assume that the distribution $\mathcal{D}$ follows a Gaussian multi-index model (Zhou et al. 2022).

(A) $d$-dimensional Gaussian features with arbitrary mean and covariance: $x \sim \mathcal{N}(\mu, \Sigma)$

(B) a generic multi-index model: there exist a low-dimensional projection $W = [w_1^*, ..., w_k^*] \in \mathbb{R}^{d \times k}$, a random variable $\xi \sim \mathcal{D}_\xi$ independent of $x$ (not necessarily Gaussian), and an unknown link function $g : \mathbb{R}^{k+1} \to \mathcal{Y}$ such that

$$\eta_i = \langle w_i^*, x \rangle, \quad y = g(\eta_1, ..., \eta_k, \xi). \quad (2)$$

We require $x$ to be Gaussian because the proof of our generalization bound depends on the Gaussian Minimax Theorem (Gordon 1985; Thrampoulidis et al. 2014), as in many other works (Koehler et al. 2021; Zhou et al. 2022; Wang et al. 2021; Donhauser et al. 2022). In settings where the features are not Gaussian, many works (Goldt et al. 2020; Mei and Montanari 2022; Misiakiewicz 2022; Hu and Lu 2023; Han and Shen 2022) have established universality results showing that the test and training error are asymptotically equivalent to the error of a Gaussian model with matching covariance matrix. However, we show in section 7 that Gaussian universality is not always valid and we discuss potential extensions of the Gaussian feature assumption there.

The multi-index assumption is a generalization of a well-specified linear model $y = \langle w^*, x \rangle + \xi$, which corresponds to $k = 1$ and $g(\eta, \xi) = \eta + \xi$ in (2). Since $g$ is not even required to be continuous, the assumption on $y$ is quite general. It allows nonlinear trend and heteroskedasticity, which pose significant challenges for the analyses using random matrix theory (but not for uniform convergence). In Zhou et al. (2022), the range of $g$ is taken to be $\mathcal{Y} = \{-1, 1\}$ in order to study binary classification. In section 5.2, we allow the conditional distribution of $y$ to have a point mass at zero, which is crucial for distinguishing ReLU regression from standard linear regression.

## 4   Optimistic Rate

In order to establish uniform convergence, we need two additional technical assumptions because we have not made any assumption on $f$ and $g$.

(C) hypercontractivity: there exists a universal constant $\tau > 0$ such that uniformly over all $(w, b) \in \mathbb{R}^d \times \mathbb{R}$, it holds that

$$\frac{\mathbb{E}[f(\langle w, x \rangle + b, y)^4]^{1/4}}{\mathbb{E}[f(\langle w, x \rangle + b, y)]} \leq \tau. \tag{3}$$

(D) the class of functions on $\mathbb{R}^{k+1} \times \mathcal{Y}$ defined below has VC dimension at most $h$:

$$\{(x, y) \rightarrow \mathbb{1}\{f(\langle w, x \rangle, y) > t\} : (w, t) \in \mathbb{R}^{k+1} \times \mathbb{R}\}. \tag{4}$$

Crucially, the quantities $\tau$ and $h$ only depend on the number of indices $k$ in assumption (B) instead of the feature dimension $d$. This is a key distinction because $k$ can be a small constant even when the feature dimension is very large. For example, recall that $k = 1$ in a well-specified model and it is completely free of $d$. The feature dimension plays no role in equation (3) because conditioned on $W^T x \in \mathbb{R}^k$ and $\xi \in \mathbb{R}$, the response $y$ is non-random and the distribution of $\langle w, x \rangle$ for any $w \in \mathbb{R}^d$ only depends on $w^T \mu$ and $w^T \Sigma w$ by properties of the Gaussian distribution.

The hypercontractivity (or "norm equivalence") assumption (3) is one of a few different common assumptions made in the statistical learning theory literature to rule out the possibility of a very heavy-tailed loss function. The reason is that if the loss function has a very tiny chance of being extremely large, it is not possible to state any meaningful learning guarantee from a finite number of samples. Hypercontractivity was used as an assumption already in Vapnik 1982. Another common and incomparable assumption made in the literature is boundedness, but this is not suitable for us because, e.g., the squared loss under the Gaussian distribution will not be bounded almost surely. On the other hand, if $f$ is the squared loss then (3) holds with $\tau = \sqrt[4]{105}$ by standard properties of the Gaussian distribution. We can also state other versions of our main theorem by using other results in statistical learning theory to handle the low-dimensional concentration; in our proof we simply use a result from Vapnik 1982 in a contained and black-box fashion. (See e.g. Mendelson 2017; Vapnik 1982 for more discussion of the different possible assumptions.)

Similar assumptions are made in Zhou et al. (2022) and more background on these assumptions can be found there, but we note that our assumption (C) and (D) are weaker than theirs because we do not require (3) and (4) to hold uniformly over all Moreau envelopes of $f$.

We are now ready to state the uniform convergence results for square-root Lipschitz losses.

**Theorem 1.** *Assume that (A), (B), (C), and (D) holds, and let $Q = I - W(W^T \Sigma W)^{-1} W^T \Sigma$. For any $\delta \in (0, 1)$, let $C_\delta : \mathbb{R}^d \rightarrow [0, \infty]$ be a continuous function such that with probability at least $1 - \delta/4$ over $x \sim \mathcal{N}(0, \Sigma)$, uniformly over all $w \in \mathbb{R}^d$,*

$$\langle w, Q^T x \rangle \leq C_\delta(w). \tag{5}$$

*If for each $y \in \mathcal{Y}$, $f$ is non-negative and $\sqrt{f}$ is $\sqrt{H}$-Lipschitz with respect to the first argument, then with probability at least $1 - \delta$, it holds that uniformly over all $(w, b) \in \mathbb{R}^d \times \mathbb{R}$, we have*

$$(1 - \epsilon) L_f(w, b) \leq \left( \sqrt{\hat{L}_f(w, b)} + \sqrt{\frac{H C_\delta(w)^2}{n}} \right)^2 \tag{6}$$

*where $\epsilon = O\left( \tau \sqrt{\frac{h \log(n/h) + \log(1/\delta)}{n}} \right)$.*

Since the label $y$ only depends on $x$ through $W^T x$, the matrix $Q$ is simply a (potentially oblique) projection such that $Q^T x$ is independent of $y$. In the proof, we separate $x$ into a low-dimensional component related to $y$ and the independent component $Q^T x$. We establish a low-dimensional concentration result using VC theory, which is reflected in the $\epsilon$ term that does not depend on $d$, and we control the the remaining high-dimensional component using a scale-sensitive measure $C_\delta$.

The complexity term $C_\delta(w)/\sqrt{n}$ should be thought of as a (localized form of) Rademacher complexity $\mathcal{R}_n$. For example, for any norm $\| \cdot \|$, we have $\langle w, Q^T x \rangle \leq \|w\| \cdot \|Q^T x\|_*$ and the Rademacher complexity for linear predictors with norm bounded by $B$ is $B \cdot \mathbb{E}\|x\|_*/\sqrt{n}$ (Shalev-Shwartz and Ben-David 2014). More generally, if we only care about linear predictors in a set $\mathcal{K}$, then $C_\delta \approx \mathbb{E}\left[ \sup_{w \in \mathcal{K}} \langle w, Q^T x \rangle \right]$ is simply the Gaussian width (Bandeira 2016; Gordon 1988; Koehler et al. 2021) of $\mathcal{K}$ with respect to $\Sigma^\perp = Q^T \Sigma Q$ and is exactly equivalent to the expected Rademacher complexity of the class of functions (e.g., Proposition 1 of Zhou et al. 2021).

# 5 Applications

We now go on to analyze several different problems using optimistic rates and sqrt-lipschitz losses. One of the key insights we want to highlight is the *identification of the consistent loss* for many of these problems — in other word, the test loss which is implicitly being minimized. When we consider interpolators/overfit models, this is not obvious because an interpolator simultaneously minimizes many different training losses (and when we look at the test loss, each different loss functional may correspond to a different minimizer). Nevertheless, we are able to analyze phenomena like benign overfitting by identifying the particular sqrt-Lipschitz losses which the interpolator is consistent in, i.e. for which it approaches the minimum of the test loss in the appropriate limit.

## 5.1 Benign Overfitting in Phase Retrieval

In this section, we study the problem of phase retrieval where only the magnitude of the label $y$ can be measured. If the number of observations $n < d$, the minimal norm interpolant is

$$\hat{w} = \underset{w \in \mathbb{R}^d : \forall i \in [n], \langle w, x_i \rangle^2 = y_i^2}{\arg\min} \|w\|_2. \tag{7}$$

We are particularly interested in the generalization error of (7) when $y_i$ are noisy labels and so $\hat{w}$ overfits to the training set. It is well-known that gradient descent initialized at zero converges to a KKT point of the problem (7) when it reaches a zero training error solution (see e.g. Gunasekar et al. 2018). While is not always computationally feasible to compute $\hat{w}$, we can study it as a theoretical model for benign overfitting in phase retrieval.

Due to our uniform convergence guarantee in Theorem 1, it suffices to analyze the norm of $\hat{w}$ and we can use the norm calculation to find the appropriate loss for phase retrieval. The key observation is that for any $w^\sharp \in \mathbb{R}^d$, we can let $I = \{i \in [n] : \langle w^\sharp, x_i \rangle \geq 0\}$ and the predictor $w = w^\sharp + w^\perp$ satisfies $|\langle w, x_i \rangle| = y_i$ where

$$w^\perp = \underset{\substack{w \in \mathbb{R}^d : \\ \forall i \in I, \langle w, x_i \rangle = y_i - |\langle w^\sharp, x_i \rangle| \\ \forall i \notin I, \langle w, x_i \rangle = |\langle w^\sharp, x_i \rangle| - y_i}}{\arg\min} \|w\|_2. \tag{8}$$

Then it holds that $\|\hat{w}\|_2 \leq \|w\|_2 \leq \|w^\sharp\|_2 + \|w^\perp\|_2$. By treating $y_i - |\langle w^\sharp, x_i \rangle|$ and $|\langle w^\sharp, x_i \rangle| - y_i$ as the residuals, the norm calculation for $\|w^\perp\|_2$ is the same as the case of linear regression (Koehler et al. 2021; Zhou et al. 2022). Going through this analysis yields the following norm bound (here $R(\Sigma) = \mathrm{Tr}(\Sigma)^2 / \mathrm{Tr}(\Sigma^2)$ is a measure of effective rank, as in (Bartlett et al. 2020)):

**Theorem 2.** *Under assumptions (A) and (B), let $f : \mathbb{R} \times \mathcal{Y} \to \mathbb{R}$ be given by $f(\hat{y}, y) := (|\hat{y}| - y)^2$ with $\mathcal{Y} = \mathbb{R}_{\geq 0}$. Let $Q$ be the same as in Theorem 1 and $\Sigma^\perp = Q^T \Sigma Q$. Fix any $w^\sharp \in \mathbb{R}^d$ such that $Qw^\sharp = 0$ and for some $\rho \in (0,1)$, it holds that*

$$\hat{L}_f(w^\sharp) \leq (1 + \rho) L_f(w^\sharp). \tag{9}$$

*Then with probability at least $1 - \delta$, for some $\epsilon \lesssim \rho + \log\left(\frac{1}{\delta}\right)\left(\frac{1}{\sqrt{n}} + \frac{1}{\sqrt{R(\Sigma^\perp)}} + \frac{k}{n} + \frac{n}{R(\Sigma^\perp)}\right)$, it holds that*

$$\min_{\substack{w \in \mathbb{R}^d : \\ \forall i \in [n], \langle w, x_i \rangle^2 = y_i^2}} \|w\|_2 \leq \|w^\sharp\|_2 + (1 + \epsilon)\sqrt{\frac{nL_f(w^\sharp)}{\mathrm{Tr}(\Sigma^\perp)}}. \tag{10}$$

Note that in this result we used the loss $f(\hat{y}, y) = (|\hat{y}| - y)^2$, which is 1-square-root Lipschitz and naturally arises in the analysis. We will now show that this is the *correct* loss in the sense that benign overfitting is consistent with this loss. To establish consistency result, we can pick $w^\sharp$ to be the minimizer of the population error. The population minimizer always satisfies $Qw^\sharp = 0$ because $L_f((I - Q)w) \leq L_f(w)$ for all $w \in \mathbb{R}^d$ by Jensen's inequality. Condition (9) can also be easily checked because we just need concentration of the empirical loss at a single non-random parameter. Applying Theorem 1, we obtain the following.

**Corollary 1.** *In the same setting as in Theorem 2, with probability at least $1 - \delta$, it holds that for some*

$$\rho \lesssim \tau \sqrt{\frac{k \log(n/k) + \log(1/\delta)}{n}} + \log\left(1/\delta\right) \left( \frac{1}{\sqrt{n}} + \frac{1}{\sqrt{R(\Sigma^\perp)}} + \frac{k}{n} + \frac{n}{R(\Sigma^\perp)} \right),$$

*the minimal norm interpolant $\hat{w}$ given by (7) enjoys the learning guarantee:*

$$L(\hat{w}, \hat{b}) \leq (1 + \rho) \left( \sqrt{L(w^\sharp, b^\sharp)} + \|w^\sharp\|_2 \sqrt{\frac{\text{Tr}(\Sigma^\perp)}{n}} \right)^2. \tag{11}$$

Therefore, $\hat{w}$ is consistent if the same benign overfitting conditions from Bartlett et al. (2020) and Zhou et al. (2022) hold: $k = o(n)$, $R(\Sigma^\perp) = \omega(n)$ and $\|w^\sharp\|_2^2 \, \text{Tr}(\Sigma^\perp) = o(n)$.

## 5.2 Benign Overfitting in ReLU Regression

Since we only need $\sqrt{f}$ to be Lipschitz, for any 1-Lipschitz function $\sigma : \mathbb{R} \to \mathbb{R}$, we can consider

- (i) $f(\hat{y}, y) = (\sigma(\hat{y}) - y)^2$ when $\mathcal{Y} = \mathbb{R}$
- (ii) $f(\hat{y}, y) = (1 - \sigma(\hat{y})y)_+^2$ when $\mathcal{Y} = \{-1, 1\}$.

We can interpret the above loss function $f$ as learning a neural network with a single hidden unit. Indeed, Theorem 1 can be straightforwardly applied to these situations. However, we do not always expect benign overfitting under these loss functions for the following simple reason, as pointed out in Shamir (2022): when $\sigma$ is invertible, interpolating the loss $f(\hat{y}, y) = (\sigma(\hat{y}) - y)^2$ is the same as interpolating the loss $f(\hat{y}, y) = (\hat{y} - \sigma^{-1}(y))^2$ and so the learned function would be the minimizer of $\mathbb{E}[(\langle w, x \rangle + b - \sigma^{-1}(y))^2]$ which is typically different from the minimizer of $\mathbb{E}[(\sigma(\langle w, x \rangle + b) - y)^2]$.

The situation of ReLU regression, where $\sigma(\hat{y}) = \max\{\hat{y}, 0\}$, is more interesting because $\sigma$ is *not* invertible. In order to be able to interpolate, we must have $y \geq 0$. If $y > 0$ with probability 1, then $\sigma(\hat{y}) = y$ is the same as $\hat{y} = y$ and we are back to interpolating the square loss $f(\hat{y}, y) = (\hat{y} - y)^2$. From this observation, we see that $f(\hat{y}, y) = (\sigma(\hat{y}) - y)^2$ cannot be the appropriate loss for consistency even though it's 1 square-root Lipschitz. In contrast, when there is some probability mass at $y = 0$, it suffices to output any non-positive value and the minimal norm to interpolate is potentially smaller than requiring $\hat{y} = 0$. Similar to the previous section, we can let $I = \{i \in [n] : y_i > 0\}$ and for any $(w^\sharp, b^\sharp) \in \mathbb{R}^{d+1}$, the predictor $w = w^\sharp + w^\perp$ satisfies $\sigma(\langle w, x_i \rangle + b^\sharp) = y_i$ where

$$w^\perp = \underset{\substack{w \in \mathbb{R}^d: \\ \forall i \in I, \langle w, x_i \rangle = y_i - \langle w^\sharp, x_i \rangle - b^\sharp \\ \forall i \notin I, \langle w, x_i \rangle = -\sigma(\langle w^\sharp, x_i \rangle + b^\sharp)}}{\arg\min} \|w\|_2 \tag{12}$$

Our analysis will show that the consistent loss for benign overfitting with ReLU regression is

$$f(\hat{y}, y) = \begin{cases} (\hat{y} - y)^2 & \text{if} \quad y > 0 \\ \sigma(\hat{y})^2 & \text{if} \quad y = 0. \end{cases} \tag{13}$$

We first state the norm bound below.

**Theorem 3.** *Under assumptions (A) and (B), let $f : \mathbb{R} \times \mathcal{Y} \to \mathbb{R}$ be the loss defined in (13) with $\mathcal{Y} = \mathbb{R}_{\geq 0}$. Let $Q$ be the same as in Theorem 1 and $\Sigma^\perp = Q^T \Sigma Q$. Fix any $(w^\sharp, b^\sharp) \in \mathbb{R}^{d+1}$ such that $Qw^\sharp = 0$ and for some $\rho \in (0, 1)$, it holds that*

$$\hat{L}_f(w^\sharp, b^\sharp) \leq (1 + \rho) L_f(w^\sharp, b^\sharp). \tag{14}$$

*Then with probability at least $1 - \delta$, for some $\epsilon \lesssim \rho + \log\left(\frac{1}{\delta}\right) \left( \frac{1}{\sqrt{n}} + \frac{1}{\sqrt{R(\Sigma^\perp)}} + \frac{k}{n} + \frac{n}{R(\Sigma^\perp)} \right)$, it holds that*

$$\underset{\substack{(w,b) \in \mathbb{R}^{d+1}: \\ \forall i \in [n], \sigma(\langle w, x_i \rangle + b) = y_i}}{\min} \|w\|_2 \leq \|w^\sharp\|_2 + (1 + \epsilon) \sqrt{\frac{n L_f(w^\sharp, b^\sharp)}{\text{Tr}(\Sigma^\perp)}}. \tag{15}$$

Since the loss defined in (13) is also 1-square-root Lipschitz, it can be straightforwardly combined with Theorem 1 to establish benign overfitting in ReLU regression under this loss. The details are exactly identical to the previous section and so we omit it here.

## 5.3 Benign Overfitting in Matrix Sensing

We now consider the problem of matrix sensing: given random matrices $A_1, ..., A_n$ (with i.i.d. standard Gaussian entries) and independent linear measurements $y_1, ..., y_n$ given by $y_i = \langle A_i, X^* \rangle + \xi_i$ where $\xi_i$ is independent of $A_i$, and $\mathbb{E}\xi = 0$ and $\mathbb{E}\xi^2 = \sigma^2$, we hope to reconstruct the matrix $X^* \in \mathbb{R}^{d_1 \times d_2}$ with sample size $n \ll d_1 d_2$. To this end, we assume that $X^*$ has rank $r$. In this setting, since the measurement matrices have i.i.d. standard Gaussian entries, the test error is closely related to the estimation error:

$$L(X) = \mathbb{E}(\langle A, X \rangle - y)^2 = \|X - X^*\|_F^2 + \sigma^2.$$

The classical approach to this problem is to find the minimum nuclear norm solution:

$$\hat{X} = \underset{X \in \mathbb{R}^{d_1 \times d_2}: \langle A_i, X \rangle = y_i}{\arg\min} \|X\|_*. \tag{16}$$

Gunasekar et al. (2017) also shows that gradient descent converges to the minimal nuclear norm solution in matrix factorization problems. It is well known that having low nuclear norm can ensure generalization (Foygel and Srebro 2011; Srebro and Shraibman 2005) and minimizing the nuclear norm ensures reconstruction (Candès and Recht 2009; Recht et al. 2010). However, if the noise level $\sigma$ is high, then even the minimal nuclear norm solution $\hat{X}$ can have large nuclear norm. Since our result can be adapted to different norms as regularizer, our uniform convergence guarantee can be directly applied. The dual norm of the nuclear norm is the spectral norm, and it is well-known that the spectrum norm of a Gaussian random matrix is approximately $\sqrt{d_1} + \sqrt{d_2}$ (Vershynin 2018). It remains to analyze the minimal nuclear norm required to interpolate. For simplicity, we assume that $\xi$ are Gaussian below, but we can extend it to be sub-Gaussian.

**Theorem 4.** *Suppose that $d_1 d_2 > n$, then there exists some $\epsilon \lesssim \sqrt{\frac{\log(32/\delta)}{n}} + \frac{n}{d_1 d_2}$ such that with probability at least $1 - \delta$, it holds that*

$$\min_{\forall i \in [n], \langle A_i, X \rangle = y_i} \|X\|_* \leq \sqrt{r}\|X^*\|_F + (1 + \epsilon)\sqrt{\frac{n\sigma^2}{d_1 \vee d_2}}. \tag{17}$$

Without loss of generality, we will assume $d_1 \leq d_2$ from now on because otherwise we can take the transpose of $A$ and $X$. Similar to assuming $n/R(\Sigma^\perp) \to 0$ in linear regression, we implicitly assume that $n/d_1 d_2 \to 0$ in matrix sensing. Such scaling is necessary for benign overfitting because of the lower bound on the test error for *any* interpolant (e.g., Proposition 4.3 of Zhou et al. 2020). Finally, we apply the uniform convergence guarantee.

**Theorem 5.** *Fix any $\delta \in (0, 1)$. There exist constants $c_1, c_2, c_3 > 0$ such that if $d_1 d_2 > c_1 n$, $d_2 > c_2 d_1$, $n > c_3 r(d_1 + d_2)$, then with probability at least $1 - \delta$ that*

$$\frac{\|\hat{X} - X^*\|_F^2}{\|X^*\|_F^2} \lesssim \frac{r(d_1 + d_2)}{n} + \sqrt{\frac{r(d_1 + d_2)}{n}} \frac{\sigma}{\|X^*\|_F} + \left(\sqrt{\frac{d_1}{d_2}} + \frac{n}{d_1 d_2}\right) \frac{\sigma^2}{\|X^*\|_F^2}. \tag{18}$$

From Theorem 5, we see that when the signal to noise ratio $\frac{\|X^*\|_F^2}{\sigma^2}$ is bounded away from zero, then we obtain consistency $\frac{\|\hat{X} - X^*\|_F^2}{\|X^*\|_F^2} \to 0$ if (i) $r(d_1 + d_2) = o(n)$, (ii) $d_1 d_2 = \omega(n)$, and (iii) $d_1/d_2 \to \{0, \infty\}$. This can happen for example when $r = \Theta(1), d_1 = \Theta(n^{1/2}), d_2 = \Theta(n^{2/3})$. As discussed earlier, the second condition is necessary for benign overfitting, and the first consistency condition should be necessary even for regularized estimators. It is possible that the final condition is not necessary and we leave a tighter understanding of matrix sensing as future work.

## 6 Single-Index Neural Networks

We show that our results extend a even more general setting than (1). Suppose that we have a parameter space $\Theta \subseteq \mathbb{R}^p$ and a continuous mapping $w$ from $\theta \in \Theta$ to a linear predictor $w(\theta) \in \mathbb{R}^d$. Given a function $f : \mathbb{R} \times \mathcal{Y} \times \Theta \to \mathbb{R}$ and i.i.d. sample pairs $(x_i, y_i)$ drawn from distribution $\mathcal{D}$, we consider training and test error of the form:

$$\hat{L}(\theta) = \frac{1}{n} \sum_{i=1}^n f(\langle w(\theta), x_i \rangle, y_i, \theta) \quad \text{and} \quad L(\theta) = \mathbb{E}[f(\langle w(\theta), x \rangle, y, \theta)]. \tag{19}$$

We make the same assumptions (A) and (B) on $\mathcal{D}$. Assumptions (C) and (D) can be naturally extended to the following:

(E) there exists a universal constant $\tau > 0$ such that uniformly over all $\theta \in \Theta$, it holds that

$$\frac{\mathbb{E}[f(\langle w(\theta), x \rangle, y, \theta)^4]^{1/4}}{\mathbb{E}[f(\langle w(\theta), x \rangle, y, \theta)]} \leq \tau. \tag{20}$$

(F) bounded VC dimensions: the class of functions on $\mathbb{R}^{k+1} \times \mathcal{Y}$ defined by

$$\{(x, y) \rightarrow \mathbb{1}\{f(\langle w, x \rangle, y, \theta) > t\} : (w, t, \theta) \in \mathbb{R}^{k+1} \times \mathbb{R} \times \Theta\} \tag{21}$$

has VC dimension at most $h$.

Now we can state the extension of Theorem 1.

**Theorem 6.** *Suppose that assumptions (A), (B), (E) and (F) hold. For any $\delta \in (0, 1)$, let $C_\delta : \mathbb{R}^d \rightarrow [0, \infty]$ be a continuous function such that with probability at least $1 - \delta/4$ over $x \sim \mathcal{N}(0, \Sigma)$, uniformly over all $\theta \in \Theta$,*

$$\langle w(\theta), Q^T x \rangle \leq C_\delta(w(\theta)). \tag{22}$$

*If for each $\theta \in \Theta$ and $y \in \mathcal{Y}$, $f$ is non-negative and $\sqrt{f}$ is $\sqrt{H_\theta}$-Lipschitz with respect to the first argument, and $H_\theta$ is continuous in $\theta$, then with probability at least $1 - \delta$, it holds that uniformly over all $\theta \in \Theta$, we have*

$$(1 - \epsilon) L(\theta) \leq \left( \sqrt{\hat{L}(\theta)} + \sqrt{\frac{H_\theta \, C_\delta(w(\theta))^2}{n}} \right)^2. \tag{23}$$

Next, we show that the generality of our result allows us to establish uniform convergence bound for two-layer neural networks with weight sharing. In particular, we let $\sigma(x) = \max(x, 0)$ be the ReLU activation function and $\theta = (w, a, b) \in \mathbb{R}^{d+2N}$, where $N$ is the number of hidden units. Consider the loss function

$$f(\hat{y}, y, \theta) = \left( \sum_{i=1}^{N} a_i \sigma(\hat{y} - b_i) - y \right)^2 \quad \text{or} \quad f(\hat{y}, y, \theta) = \left( 1 - \sum_{i=1}^{N} a_i \sigma(\hat{y} - b_i) y \right)_+^2, \tag{24}$$

then $L(\theta)$ is the test error of a neural network of the form $h_\theta(x) := \sum_{i=1}^{N} a_i \sigma(\langle w, x \rangle - b_i)$. Since our uniform convergence guarantee holds uniformly over all $\theta$, it applies to networks whose first and second layer weights are optimized at the same time. Without loss of generality, we can assume that $b_1 \leq ... \leq b_N$ are sorted, then it is easy to see that $\sqrt{f}$ is $\max_{j \in [N]} \left| \sum_{i=1}^{j} a_i \right|$ Lipschitz. Applying Theorem 6, we obtain the following corollary.

**Corollary 2.** *Fix an arbitrary norm $\| \cdot \|$ and consider $f$ as defined in (24). Assume that the data distribution $\mathcal{D}$ satisfy (A), (B), and (E). Then with probability at least $1 - \delta$, it holds that uniformly over all $\theta = (w, a, b) \in \mathbb{R}^{d+2N}$, we have*

$$(1 - \epsilon) L(\theta) \leq \left( \sqrt{\hat{L}(\theta)} + \frac{\max_{j \in [N]} \left| \sum_{i=1}^{j} a_i \right| \|w\| \left( \mathbb{E}\|Q^T x\|_* + \epsilon' \right)}{\sqrt{n}} \right)^2 \tag{25}$$

*where $\epsilon$ is the same as in Theorem 6 with $h = \tilde{O}(k + N)$ and $\epsilon' = O\left( \sup_{\|u\| \leq 1} \|u\|_{\Sigma^\perp} \sqrt{\log(1/\delta)} \right)$.*

The above theorem says given a network parameter $\theta$, after sorting the $b_i$'s, a good complexity measure to look at is $\max_{j \in [N]} \left| \sum_{i=1}^{j} a_i \right| \cdot \|w\|$ and equation (25) precisely quantify how the complexity of a network controls generalization.

## 7 On Gaussian Universality

In this section, we discuss a counterexample to Gaussian universality motivated by Shamir (2022). In particular, we consider a data distribution $\mathcal{D}$ over $(x, y)$ given by:

(G) $x = (x_{|k}, x_{|d-k})$ where $x_{|k} \sim \mathcal{N}(0, \Sigma_{|k})$ and there exists a function $h : \mathbb{R}^k \to \mathbb{R}$ such that

$$x_{|d-k} = h(x_{|k}) \cdot z \quad \text{with} \quad z \sim \mathcal{N}\left(0, \Sigma_{|d-k}\right) \quad \text{independent of } x_{|k} \qquad (26)$$

(H) there exists a function $g : \mathbb{R}^{k+1} \to \mathbb{R}$ such that

$$y = g(x_{|k}, \xi) \qquad (27)$$

where $\xi \sim \mathcal{D}_\xi$ is independent of $x$ (but not necessarily Gaussian).

By the independence between $z$ and $x_{|k}$, we can easily see that $x_{|k}$ and $x_{|d-k}$ are uncorrelated. Moreover, the covariance matrix of $x_{|d-k}$ is

$$\mathbb{E}[x_{|d-k} x_{|d-k}^T] = \mathbb{E}[h(x_{|k})^2] \cdot \Sigma_{|d-k}$$

which is just a re-scaling of $\Sigma_{|d-k}$ and therefore has the same effective rank as $R(\Sigma_{|d-k})$. Even though the feature $x$ in $\mathcal{D}$ is non-Gaussian, its tail $x_{|d-k}$ is Gaussian conditioned on $x_{|k}$. Therefore, our proof technique is still applicable after a conditioning step. However, it turns out that the uniform convergence bound in Theorem 1 is no longer valid. Instead, we have to re-weight the loss function. The precise theorem statements can be found in the appendix. We briefly describe our theoretical results below, which follow the same ideas as in section 4 and 5.

**Uniform Convergence.** Under a similar low-dimensional concentration assumption such as (C), if we let $C_\delta$ be such that $\langle w, z \rangle \leq C_\delta(w)$ for all $w \in \mathbb{R}^{d-k}$ with high probability, it holds that

$$\mathbb{E}\left[\frac{f(\langle w, x \rangle, y)}{h(x_{|k})^2}\right] \leq (1 + o(1))\left(\frac{1}{n}\sum_{i=1}^{n}\frac{f(\langle w, x_i \rangle, y_i)}{h(x_{i|k})^2} + \frac{C_\delta(w_{|d-k})}{\sqrt{n}}\right)^2. \qquad (28)$$

Note that the distribution of $z$ is specified in (G) and different to the distribution of $x_{|d-k}$.

**Norm Bound.** Next, we focus on linear regression and compute the minimal norm required to interpolate. If we pick $\Sigma_{|d-k}$ to satisfy the benign overfitting conditions, for any $w_{|k}^* \in \mathbb{R}^k$, we have

$$\min_{\substack{w \in \mathbb{R}^d: \\ \forall i \in [n], \langle w, x_i \rangle = y_i}} \|w\|_2^2 \leq \|w_{|k}^*\|_2^2 + (1 + o(1))\frac{n\mathbb{E}\left[\left(\frac{y - \langle w_{|k}^*, x_{|k} \rangle}{h(x_{|k})}\right)^2\right]}{\operatorname{Tr}(\Sigma_{|d-k})} \qquad (29)$$

It is easy to see that the $w^*$ that minimizes the population weighted square loss satisfy $w_{|d-k}^* = 0$, and so we can let $w_{|k}^*$ to be the minimizer.

**Benign Overfitting.** Let $\hat{w} = \arg\min_{w \in \mathbb{R}^d: Xw=Y}\|w\|_2$ be the minimal norm interpolant. Plugging in the norm bound (29) into the uniform convergence guarantee (28), we show that

$$\mathbb{E}\left[\frac{(\langle \hat{w}, x \rangle - y)^2}{h(x_{|k})^2}\right] \leq (1 + o(1))\frac{\|\hat{w}\|_2^2 \operatorname{Tr}(\Sigma_{|d-k})}{n} \to \mathbb{E}\left[\frac{(\langle w^*, x \rangle - y)^2}{h(x_{|k})^2}\right]. \qquad (30)$$

which recovers the consistency result of Shamir (2022).

**Contradiction.** Suppose that Gaussian universality holds, then our optimistic rate theory would predict that

$$\mathbb{E}[(\langle \hat{w}, x \rangle - y)^2] \leq (1 + o(1))\frac{\|\hat{w}\|_2^2 \cdot \mathbb{E}[h(x_{|k})^2]\operatorname{Tr}(\Sigma_{|d-k})}{n}. \qquad (31)$$

Combining (30) with (31), we obtain that

$$\min_w \mathbb{E}[(\langle w, x \rangle - y)^2] \leq (1 + o(1))\min_w \mathbb{E}[h(x_{|k})^2] \cdot \mathbb{E}\left[\left(\frac{y - \langle w, x \rangle}{h(x_{|k})}\right)^2\right] \qquad (32)$$

which cannot always hold. For example, let's consider the case where $k = 1$, $x_1 \sim \mathcal{N}(0, 1)$, $h(x_1) = 1 + |x_1|$ and $y = h(x_1)^2$. Then it is straightforward to verify that the left hand side of (32) equals $\mathbb{E}[h(x_1)^4]$ and the right hand side equals $\mathbb{E}[h(x_1)^2]^2$, but this is impossible because

$$\mathbb{E}[h(x_1)^4] - \mathbb{E}[h(x_1)^2]^2 = \mathrm{Var}(h(x_1)^2) > 0.$$

In the counterexample above, we see that it is possible to introduce strong dependence between the signal and the tail component of $x$ while ensuring that they are uncorrelated. The dependence will prevent the norm of the tail from concentrating around its mean, no matter how large the sample size is. In contrast, the norm of the tail will concentrate for Gaussian features with a matching covariance — such discrepancy results in an over-optimistic bound for non-Gaussian data.

## 8 Conclusion

In this paper, we extend a type of sharp uniform convergence guarantee proven for the square loss in Zhou et al. (2021) to any *square-root Lipschitz* loss. Uniform convergence with square-root Lipschitz loss is an important tool because the appropriate loss to study interpolation learning is usually square-root Lipschitz instead of Lipschitz. Compared to the prior work of Zhou et al. 2022, our view significantly simplify the assumptions to establish optimistic rate. Since we don't need to explicitly compute the Moreau envelope for each application, our framework easily leads to many novel benign overfitting results, including low-rank matrix sensing.

In the applications to phase retrieval and ReLU, we identify the appropriate loss function $f$ and our norm calculation overcomes the challenge that $f$ is non-convex and so CGMT cannot be directly applied. Furthermore, we explore new extensions of the uniform convergence technique to study single-index neural networks and suggest a promising research direction to understand the generalization of neural networks. Finally, we argue that Gaussian universality cannot always be taken for granted by analyzing a model where only the weighted square loss enjoys an optimistic rate. Our results highlight the importance of tail concentration and shed new lights on the necessary conditions for universality. An important future direction is to extend optimistic rate beyond Gaussian data, possibly through worst-case Rademacher complexity. Understanding the performance of more practical algorithms in phase retrieval and deriving the necessary and sufficient conditions for benign overfitting in matrix sensing are also interesting problems.

**Acknowledgements.** F.K. was supported in part by NSF award CCF-1704417, NSF award IIS-1908774, and N. Anari's Sloan Research Fellowship.

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

# A  Organization of the Appendices

In the Appendix, we give proofs of all results from the main text. In Appendix B, we study properties of square-root-Lipschitz functions and introduce some technical tools that we use throughout the appendix. In Appendix C, we prove our main uniform convergence guarantee (Theorem 1 and the more general version Theorem 6). In Appendix D, we obtain bounds on the minimal norm required to interpolate in the settings studied in section 5. In Appendix E, we provide details on the counterexample to Gaussian universality described in section 7.

# B  Preliminaries

## B.1  Properties of Square-root Lipschitz Loss

In this section, we prove that square-root Lipschitzness can be equivalently characterized by a relationship between a function and its Moreau envelope, which can be used to establish uniform convergence results based on the recent work of Zhou et al. 2022. We formally define Lipschitz functions and Moreau envelope below.

**Definition 1.** A function $f : \mathbb{R} \to \mathbb{R}$ is $M$-Lipschitz if for all $x, y$ in $\mathbb{R}$,

$$|f(x) - f(y)| \leq M|x - y|. \tag{33}$$

**Definition 2.** The Moreau envelope of a function $f : \mathbb{R} \to \mathbb{R}$ associated with smoothing parameter $\lambda \in \mathbb{R}_+$ is defined as

$$f_\lambda(x) := \inf_{y \in \mathbb{R}} f(y) + \lambda(y - x)^2. \tag{34}$$

Though we define Lipschitz functions and Moreau envelope for univariate functions from $\mathbb{R}$ to $\mathbb{R}$ above, we can easily extend definitions 1 and 2 to loss functions $f : \mathbb{R} \times \mathcal{Y} \to \mathbb{R}$ or $f : \mathbb{R} \times \mathcal{Y} \times \Theta \to \mathbb{R}$. We say a function $f : \mathbb{R} \times \mathcal{Y} \to \mathbb{R}$ is $M$-Lipschitz if for any $y \in \mathcal{Y}$ and $\hat{y}_1, \hat{y}_2 \in \mathbb{R}$, we have

$$|f(\hat{y}_1, y) - f(\hat{y}_2, y)| \leq M|\hat{y}_1 - \hat{y}_2|.$$

Similarly, we say a function $f : \mathbb{R} \times \mathcal{Y} \times \Theta \to \mathbb{R}$ is $M$-Lipschitz if for any $y \in \mathcal{Y}, \theta \in \Theta$ and $\hat{y}_1, \hat{y}_2 \in \mathbb{R}$, we have

$$|f(\hat{y}_1, y, \theta) - f(\hat{y}_2, y, \theta)| \leq M|\hat{y}_1 - \hat{y}_2|.$$

We can also define the Moreau envelope of a function $f : \mathbb{R} \times \mathcal{Y} \to \mathbb{R}$ by

$$f_\lambda(\hat{y}, y) := \inf_{u \in \mathbb{R}} f(u, y) + \lambda(u - \hat{y})^2,$$

and the Moreau envelope of a function $f : \mathbb{R} \times \mathcal{Y} \times \Theta \to \mathbb{R}$ is defined as

$$f_\lambda(\hat{y}, y, \theta) := \inf_{u \in \mathbb{R}} f(u, y, \theta) + \lambda(u - \hat{y})^2.$$

The proof of all results in this section can be straightforwardly extended to these settings. For simplicity, we ignore the additional arguments in $\mathcal{Y}$ and $\Theta$ in this section.

The Moreau envelope is usually viewed as a smooth approximation to the original function $f$; its minimizer is known as the proximal operator. It plays an important role in convex analysis (see e.g. Boyd et al. 2004; Bauschke, Combettes, et al. 2011; Rockafellar 1970), but is also useful and well-defined when $f$ is nonconvex. The canonical example of a $\sqrt{H}$-square-root-Lipschitz function is $f(x) = Hx^2$, for which we can easily check

$$f_\lambda(x) = \frac{\lambda}{\lambda + H} f(x).$$

In proposition 1 below, we show that the condition $f_\lambda \geq \frac{\lambda}{\lambda+H} f$ is exactly equivalent to $\sqrt{H}$-square-root-Lipschitzness.

**Proposition 1.** *A function $f : \mathbb{R} \to \mathbb{R}$ is non-negative and $\sqrt{H}$-square-root-Lipschitz if and only if for any $x \in \mathbb{R}$ and $\lambda \geq 0$, it holds that*

$$f_\lambda(x) \geq \frac{\lambda}{\lambda + H} f(x). \tag{35}$$

*Proof.* Suppose that equation (35) holds, then by taking $\lambda = 0$ and the definition in equation (2), we see that $f$ must be non-negative. For an non-negative function $f$, we observe for any $x \in \mathbb{R}$, it holds that

$$\forall \lambda \geq 0, \ f_\lambda(x) \geq \frac{\lambda}{\lambda + H} f(x)$$

$$\iff \forall \lambda > 0, \ f_\lambda(x) \geq \frac{\lambda}{\lambda + H} f(x) \qquad \text{since } f_\lambda \geq 0$$

$$\iff \inf_{\lambda > 0} \frac{\lambda + H}{\lambda} f_\lambda(x) \geq f(x)$$

$$\iff \inf_{\lambda > 0} \frac{\lambda + H}{\lambda} \inf_{y \in \mathbb{R}} f(y) + \lambda(y - x)^2 \geq f(x) \qquad \text{by equation (2)}$$

$$\iff \inf_{y \in \mathbb{R}} \inf_{\lambda > 0} \left(1 + \frac{H}{\lambda}\right) f(y) + (\lambda + H)(y - x)^2 \geq f(x)$$

$$\iff \inf_{y \in \mathbb{R}} f(y) + H(y - x)^2 + 2\sqrt{f(y)H(y - x)^2} \geq f(x) \qquad \text{by } \lambda^* = \sqrt{\frac{Hf(y)}{(y - x)^2}}$$

$$\iff \forall y \in \mathbb{R}, \ (\sqrt{f(y)} + \sqrt{H}|y - x|)^2 \geq f(x)$$

$$\iff \forall y \in \mathbb{R}, \ \sqrt{H}|y - x| \geq \sqrt{f(x)} - \sqrt{f(y)} \qquad \text{since } f \geq 0.$$

Therefore, $f$ must be $\sqrt{H}$-square-root-Lipschitz as well. Conversely, if $f$ is non-negative and $\sqrt{H}$-square-root-Lipschitz, then the above implies that (2) must hold and we are done. $\qquad \square$

Interestingly, there is a similar equivalent characterization for Lipschitz functions as well.

**Proposition 2.** *A function $f : \mathbb{R} \to \mathbb{R}$ is $M$-Lipschitz if and only if for any $x \in \mathbb{R}$ and $\lambda > 0$, it holds that*

$$f_\lambda(x) \geq f(x) - \frac{M^2}{4\lambda}. \tag{36}$$

*Proof.* Observe that for any $x \in \mathbb{R}$, it holds that

$$\forall \lambda > 0, \ f_\lambda(x) \geq f(x) - \frac{M^2}{4\lambda}$$

$$\iff \inf_{\lambda > 0} f_\lambda(x) + \frac{M^2}{4\lambda} \geq f(x)$$

$$\iff \inf_{\lambda > 0, y \in \mathbb{R}} f(y) + \lambda(y - x)^2 + \frac{M^2}{4\lambda} \geq f(x) \qquad \text{by equation (2)}$$

$$\iff \inf_{y \in \mathbb{R}} f(y) + M|y - x| \geq f(x) \qquad \text{by } \lambda^* = \frac{M}{2|y - x|}$$

$$\iff \forall y \in \mathbb{R}, \ M|y - x| \geq f(x) - f(y)$$

and we are done. $\qquad \square$

Finally, we show that any smooth loss is square-root-Lipschitz. Therefore, the class of square-root-Lipschitz losses is more general than the class of smooth losses studied in Srebro et al. 2010.

**Definition 3.** A twice differentiable[1] function $f : \mathbb{R} \to \mathbb{R}$ is $H$-smooth if for all $x$ in $\mathbb{R}$

$$|f''(x)| \leq H.$$

The following result is similar to to Lemma 2.1 in Srebro et al. 2010:

**Proposition 3.** *Let $f : \mathbb{R} \to \mathbb{R}$ be a $H$-smooth and non-negative function. Then for any $x \in \mathbb{R}$, it holds that*

$$|f'(x)| \leq \sqrt{2Hf(x)}.$$

*Therefore, $\sqrt{f}$ is $\sqrt{H/2}$-Lipschitz.*

---

[1]The definition of smoothness can be stated without twice differentiability, by instead requiring the gradient to be Lipschitz. We make this assumption here simply for convenience.

*Proof.* Since $f$ is $H$-smooth and non-negative, by Taylor's theorem, for any $x, y \in \mathbb{R}$, we have

$$
\begin{aligned}
0 &\leq f(y) \\
&= f(x) + f'(x)(y - x) + \frac{f''(a)}{2}(y - x)^2 \\
&\leq f(x) + f'(x)(y - x) + \frac{H}{2}(y - x)^2
\end{aligned}
$$

where $a \in [\min(x, y), \max(x, y)]$. Setting $y = x - \frac{f'(x)}{H}$ yields the desired bound. To show that $\sqrt{f}$ is Lipschitz, we observe that for any $x \in \mathbb{R}$

$$
\left| \frac{d}{dx} \sqrt{f(x)} \right| = \left| \frac{f'(x)}{2\sqrt{f(x)}} \right| \leq \sqrt{H/2}
$$

and so we apply Taylor's theorem again to show that

$$
\left| \sqrt{f(x)} - \sqrt{f(y)} \right| \leq \sqrt{H/2}\, |x - y|
$$

which is the desired definition. $\qquad\square$

### B.2 Properties of Gaussian Distribution

We will make use of the following results without proof.

**Gaussian Minimax Theorem.** Our proof of Theorem 1 and 6 will closely follow prior works that apply Gaussian Minimax Theorem (GMT) to uniform convergence (Koehler et al. 2021; Zhou et al. 2021; Zhou et al. 2022; Wang et al. 2021; Donhauser et al. 2022). The following result is Theorem 3 of Thrampoulidis et al. 2015 (see also Theorem 1 in the same reference). As explained there, it is a consequence of the main result of Gordon (1985), known as Gordon's Theorem.

**Theorem 7** (Thrampoulidis et al. 2015; Gordon 1985)**.** *Let $Z : n \times d$ be a matrix with i.i.d. $\mathcal{N}(0, 1)$ entries and suppose $G \sim \mathcal{N}(0, I_n)$ and $H \sim \mathcal{N}(0, I_d)$ are independent of $Z$ and each other. Let $S_w, S_u$ be compact sets and $\psi : S_w \times S_u \to \mathbb{R}$ be an arbitrary continuous function. Define the* Primary Optimization (PO) *problem*

$$
\Phi(Z) := \min_{w \in S_w} \max_{u \in S_u} \langle u, Zw \rangle + \psi(w, u) \tag{37}
$$

*and the* Auxiliary Optimization (AO) *problem*

$$
\phi(G, H) := \min_{w \in S_w} \max_{u \in S_u} \|w\|_2 \langle G, u \rangle + \|u\|_2 \langle H, w \rangle + \psi(w, u). \tag{38}
$$

*Under these assumptions, $\Pr(\Phi(Z) < c) \leq 2 \Pr(\phi(G, H) \leq c)$ for any $c \in \mathbb{R}$.*

*Furthermore, if we suppose that $S_w, S_u$ are convex sets and $\psi(w, u)$ is convex in $w$ and concave in $u$, then $\Pr(\Phi(Z) > c) \leq 2 \Pr(\phi(G, H) \geq c)$.*

GMT is an extremely useful tool because it allows us to convert a problem involving a random matrix into a problem involving only two random vectors. In our analysis, we will make use of a slightly more general version of Theorem 7, introduced by Koehler et al. (2021), to include additional variables which only affect the deterministic term in the minmax problem.

**Theorem 8** (Variant of GMT)**.** *Let $Z : n \times d$ be a matrix with i.i.d. $\mathcal{N}(0, 1)$ entries and suppose $G \sim \mathcal{N}(0, I_n)$ and $H \sim \mathcal{N}(0, I_d)$ are independent of $Z$ and each other. Let $S_W, S_U$ be compact sets in $\mathbb{R}^d \times \mathbb{R}^{d'}$ and $\mathbb{R}^n \times \mathbb{R}^{n'}$ respectively, and let $\psi : S_W \times S_U \to \mathbb{R}$ be an arbitrary continuous function. Define the* Primary Optimization (PO) *problem*

$$
\Phi(Z) := \min_{(w, w') \in S_W} \max_{(u, u') \in S_U} \langle u, Zw \rangle + \psi((w, w'), (u, u')) \tag{39}
$$

*and the* Auxiliary Optimization (AO) *problem*

$$
\phi(G, H) := \min_{(w, w') \in S_W} \max_{(u, u') \in S_U} \|w\|_2 \langle G, u \rangle + \|u\|_2 \langle H, w \rangle + \psi((w, w'), (u, u')). \tag{40}
$$

*Under these assumptions, $\Pr(\Phi(Z) < c) \leq 2 \Pr(\phi(G, H) \leq c)$ for any $c \in \mathbb{R}$.*

Theorem 8 requires $S_W$ and $S_U$ to be compact. However, we can usually get around the compactness requirement by a truncation argument.

**Lemma 1** (Zhou et al. 2022, Lemma 6). *Let $f : \mathbb{R}^d \to \mathbb{R}$ be an arbitrary function and $\mathcal{S}_r^d = \{x \in \mathbb{R}^d : \|x\|_2 \leq r\}$, then for any set $\mathcal{K}$, it holds that*

$$\lim_{r \to \infty} \sup_{w \in \mathcal{K} \cap \mathcal{S}_r^d} f(w) = \sup_{w \in \mathcal{K}} f(w). \tag{41}$$

*If $f$ is a random function, then for any $t \in \mathbb{R}$*

$$\Pr\left(\sup_{w \in \mathcal{K}} f(w) > t\right) = \lim_{r \to \infty} \Pr\left(\sup_{w \in \mathcal{K} \cap \mathcal{S}_r^d} f(w) > t\right). \tag{42}$$

**Lemma 2** (Zhou et al. 2022, Lemma 7). *Let $\mathcal{K}$ be a compact set and $f, g$ be continuous real-valued functions on $\mathbb{R}^d$. Then it holds that*

$$\lim_{r \to \infty} \sup_{w \in \mathcal{K}} \inf_{0 \leq \lambda \leq r} \lambda f(w) + g(w) = \sup_{w \in \mathcal{K} : f(w) \geq 0} g(w). \tag{43}$$

*If $f$ and $g$ are random functions, then for any $t \in \mathbb{R}$*

$$\Pr\left(\sup_{w \in \mathcal{K} : f(w) \geq 0} g(w) \geq t\right) = \lim_{r \to \infty} \Pr\left(\sup_{w \in \mathcal{K}} \inf_{0 \leq \lambda \leq r} \lambda f(w) + g(w) \geq t\right). \tag{44}$$

**Concentration inequalities.** Let $\sigma_{\min}(A)$ denote the minimum singular value of an arbitrary matrix $A$, and $\sigma_{\max}$ the maximum singular value. We use $\|A\|_{op} = \sigma_{\max}(A)$ to denote the operator norm of matrix $A$. The following concentration results for Gaussian vector and matrix are standard.

**Lemma 3** (Special case of Theorem 3.1.1 of Vershynin 2018). *Suppose that $Z \sim \mathcal{N}(0, I_n)$. Then*

$$\Pr(\big|\|Z\|_2 - \sqrt{n}\big| \geq t) \leq 4e^{-t^2/4}. \tag{45}$$

**Lemma 4** (Koehler et al. 2021, Lemma 10). *For any covariance matrix $\Sigma$ and $H \sim \mathcal{N}(0, I_d)$, with probability at least $1 - \delta$, it holds that*

$$1 - \frac{\|\Sigma^{1/2}H\|_2^2}{\mathrm{Tr}(\Sigma)} \lesssim \frac{\log(4/\delta)}{\sqrt{R(\Sigma)}} \tag{46}$$

*and*

$$\|\Sigma H\|_2^2 \lesssim \log(4/\delta)\,\mathrm{Tr}(\Sigma^2). \tag{47}$$

*Therefore, provided that $R(\Sigma) \gtrsim \log(4/\delta)^2$, it holds that*

$$\left(\frac{\|\Sigma H\|_2}{\|\Sigma^{1/2}H\|_2}\right)^2 \lesssim \log(4/\delta)\frac{\mathrm{Tr}(\Sigma^2)}{\mathrm{Tr}(\Sigma)}. \tag{48}$$

**Theorem 9** (Vershynin 2010, Corollary 5.35). *Let $n, N \in \mathbb{N}$. Let $A \in \mathbb{R}^{N \times n}$ be a random matrix with entries i.i.d. $\mathcal{N}(0, 1)$. Then for any $t > 0$, it holds with probability at least $1 - 2\exp(-t^2/2)$ that*

$$\sqrt{N} - \sqrt{n} - t \leq \sigma_{min}(A) \leq \sigma_{max}(A) \leq \sqrt{N} + \sqrt{n} + t. \tag{49}$$

**Conditional Distribution of Gaussian.** To handle arbitrary multi-index conditional distributions of $y$ given by assumption (B), we will apply a conditioning argument. After conditioning on $W^T x$ and $\xi$, the response $y$ is no longer random. Importantly, the conditional distribution of $x$ remains Gaussian (though with a different mean and covariance) and so we can still apply GMT. In the lemma below, $Z \in \mathbb{R}^{n \times d}$ is a random matrix with i.i.d. $\mathcal{N}(0, 1)$ entries and $X = Z\Sigma^{1/2}$.

**Lemma 5** (Zhou et al. 2022, Lemma 4). *Fix any integer $k < d$ and any $k$ vectors $w_1^*, ..., w_k^*$ in $\mathbb{R}^d$ such that $\Sigma^{1/2}w_1^*, ..., \Sigma^{1/2}w_k^*$ are orthonormal. Denoting*

$$P = I_d - \sum_{i=1}^{k} (\Sigma^{1/2}w_i^*)(\Sigma^{1/2}w_i^*)^T, \tag{50}$$

*the distribution of $X$ conditional on $Xw_1^* = \eta_1, ..., Xw_k^* = \eta_k$ is the same as that of*

$$\sum_{i=1}^{k} \eta_i(\Sigma w_i^*)^T + ZP\Sigma^{1/2}. \tag{51}$$

### B.3 Vapnik-Chervonenkis (VC) theory

By the conditioning step mentioned above, we will separate $x$ into a low-dimensional component $W^T x$ and the independent component $Q^T x$. Concentration results for the low-dimensional component can be easily established using VC theory. As mentioned in Zhou et al. 2022, low-dimensional concentration can be established using alternative results (e.g., Vapnik 1982; Panchenko 2002; Panchenko 2003; Mendelson 2017).

Recall the following definition of VC-dimension from Shalev-Shwartz and Ben-David (2014).

**Definition 4.** Let $\mathcal{H}$ be a class of functions from $\mathcal{X}$ to $\{0, 1\}$ and let $C = \{c_1, ..., c_m\} \subset \mathcal{X}$. The restriction of $\mathcal{H}$ to $C$ is

$$\mathcal{H}_C = \{(h(c_1), ..., h(c_m)) : h \in \mathcal{H}\}.$$

A hypothesis class $\mathcal{H}$ *shatters* a finite set $C \subset \mathcal{X}$ if $|\mathcal{H}_C| = 2^{|C|}$. The VC-dimension of $\mathcal{H}$ is the maximal size of a set that can be shattered by $\mathcal{H}$. If $\mathcal{H}$ can shatter sets of arbitrary large size, we say $\mathcal{H}$ has infinite VC-dimension.

Also, we have the following well-known result for the class of nonhomogenous halfspaces in $\mathbb{R}^d$ (Theorem 9.3 of Shalev-Shwartz and Ben-David (2014)), and the result on VC-dimension of the union of two hypothesis classes (Lemma 3.2.3 of Blumer et al. (1989)):

**Theorem 10.** *The class $\{x \mapsto sign(\langle w, x \rangle + b) : w \in \mathbb{R}^d, b \in \mathbb{R}\}$ has VC-dimension $d + 1$.*

**Theorem 11.** *Let $\mathcal{H}$ a hypothesis classes of finite VC-dimension $d \geq 1$. Let $\mathcal{H}_2 := \{\max(h_1, h_2) : h_1, h_2 \in \mathcal{H}\}$ and $\mathcal{H}_3 := \{\min(h_1, h_2) : h_1, h_2 \in \mathcal{H}\}$. Then, both the VC-dimension of $\mathcal{H}_2$ and the VC-dimension of $\mathcal{H}_3$ are $O(d)$.*

By combining Theorem 10 and 11, we can easily verify the VC assumption in Corollary 1 for the phase retrieval loss $f(\hat{y}, y) = (|\hat{y}| - y)^2$. Similar results can be proven for ReLU regression. To verify the VC assumption for single-index neural nets in Corollary 2, we can use the following result (equation 2 of Bartlett et al. (2019)):

**Theorem 12.** *The VC-dimension of a neural network with piecewise linear activation function, $W$ parameters, and $L$ layers has VC-dimension $O(WL \log W)$.*

We can easily establish low-dimensional concentration due to the following result:

**Theorem 13** (Vapnik 1982, Special case of Assertion 4 in Chapter 7.8; see also Theorem 7.6)**.** *Suppose that the loss function $l : \mathcal{Z} \times \Theta \to \mathbb{R}_{\geq 0}$ satisfies*

*(i) for every $\theta \in \Theta$, the function $l(\cdot, \theta)$ is measurable with respect to the first argument*

*(ii) the class of functions $\{z \mapsto \mathbb{1}\{l(z, \theta) > t\} : (\theta, t) \in \Theta \times \mathbb{R}\}$ has VC-dimension at most $h$*

*and the distribution $\mathcal{D}$ over $\mathcal{Z}$ satisfies for every $\theta \in \Theta$*

$$\frac{\mathbb{E}_{z \sim \mathcal{D}}[l(z, \theta)^4]^{1/4}}{\mathbb{E}_{z \sim \mathcal{D}}[l(z, \theta)]} \leq \tau, \tag{52}$$

*then for any $n > h$, with probability at least $1 - \delta$ over the choice of $(z_1, \ldots, z_n) \sim \mathcal{D}^n$, it holds uniformly over all $\theta \in \Theta$ that*

$$\frac{1}{n} \sum_{i=1}^{n} l(z_i, \theta) \geq \left(1 - 8\tau \sqrt{\frac{h(\log(2n/h) + 1) + \log(12/\delta)}{n}}\right) \mathbb{E}_{z \sim \mathcal{D}}[l(z, \theta)]. \tag{53}$$

## C  Proof of Theorem 6

It is clear that Theorem 1 is a special case of Theorem 6. Therefore, we will prove the more general result here.

**Notation.**   Following the tradition in statistics, we denote $X = (x_1, ..., x_n)^T \in \mathbb{R}^{n \times d}$ as the design matrix. In the proof section, we slightly abuse the notation of $\eta_i$ to mean $Xw_i^*$ and $\xi$ to mean the $n$-dimensional random vector whose $i$-th component satisfies $y_i = g(\eta_{1,i}, ..., \eta_{k,i}, \xi_i)$. We will write $X = Z\Sigma^{1/2}$ where $Z$ is a random matrix with i.i.d. standard normal entries if $\mu = 0$.

Throughout this section, we can first assume $\mu = 0$ in Assumption (A) without loss of generality because if we define $\tilde{f} : \mathbb{R} \times \mathcal{Y} \times \Theta \to \mathbb{R}$ by

$$\tilde{f}(\hat{y}, y, \theta) := f(\hat{y} + \langle w(\theta), \mu \rangle, y, \theta), \tag{54}$$

then by definition, it holds that

$$f(\langle w(\theta), x \rangle, y, \theta) = \tilde{f}(\langle w(\theta), x - \mu \rangle, y, \theta)$$

and so we can apply the theory on $\tilde{f}$ first and then translate to the problem on $f$. Similarly, we can also assume $\Sigma^{1/2} w_1^*, ..., \Sigma^{1/2} w_k^*$ are orthonormal without loss of generality. This is because we can denote $W \in \mathbb{R}^{d \times k}$ by $W = [w_1^*, ..., w_k^*]$ and let $\tilde{W} = W(W^T \Sigma W)^{-1/2}$. By definition, it holds that $\tilde{W}^T \Sigma \tilde{W} = I$ and so the columns of $\tilde{W} = [\tilde{w}_1^*, ..., \tilde{w}_k^*]$ satisfy $\Sigma^{1/2} \tilde{w}_1^*, ..., \Sigma^{1/2} \tilde{w}_k^*$ are orthonormal. If we define $\tilde{g} : \mathbb{R}^{k+1} \to \mathbb{R}$ by

$$\tilde{g}(\eta_1, ..., \eta_k, \xi) = g([\eta_1, ..., \eta_k](W^T \Sigma W)^{1/2} + \mu^T W, \xi), \tag{55}$$

then $y = \tilde{g}(x^T \tilde{W}, \xi)$ and so we can apply the theory on $\tilde{g}$.

We will write the generalization problem as a Primary Optimization problem in Theorem 8. For generality, we will let $F$ be any deterministic function and then choose it in the end.

**Lemma 6.** *Fix an arbitrary set $\Theta \subseteq \mathbb{R}^p$ and let $F : \Theta \to \mathbb{R}$ be any deterministic and continuous function. Consider dataset $(X, Y)$ drawn i.i.d. from the data distribution $\mathcal{D}$ according to (A) and (B) with $\mu = 0$ and orthonormal $\Sigma^{1/2} w_1^*, ..., \Sigma^{1/2} w_k^*$. Then conditioned on $X w_1^* = \eta_1, ..., X w_k^* = \eta_k$ and $\xi$, if we define*

$$\Phi := \sup_{\substack{(w,u,\theta) \in \mathbb{R}^d \times \mathbb{R}^n \times \Theta \\ w = P\Sigma^{1/2} w(\theta)}} \inf_{\lambda \in \mathbb{R}^n} \langle \lambda, Zw \rangle + \psi(u, \theta, \lambda \mid \eta_1, ..., \eta_k, \xi) \tag{56}$$

*where $P$ is defined in (50) and $\psi$ is a deterministic and continuous function given by*

$$
\begin{aligned}
\psi(u, \theta, \lambda \mid \eta_1, ..., \eta_k, \xi) = {} & F(\theta) - \frac{1}{n} \sum_{i=1}^n f(u_i, g(\eta_{1,i}, ..., \eta_{k,i}, \xi_i), \theta) \\
& + \left\langle \lambda, \left( \sum_{i=1}^k \eta_i (\Sigma w_i^*)^T \right) w(\theta) - u \right\rangle,
\end{aligned}
\tag{57}
$$

*then it holds that for any $t \in \mathbb{R}$, we have*

$$\Pr\left( \sup_{\theta \in \Theta} F(\theta) - \hat{L}(\theta) > t \,\Big|\, \eta_1, ..., \eta_k, \xi \right) = \Pr(\Phi > t). \tag{58}$$

*Proof.* By introducing a variable $u = Xw(\theta)$, we have

$$\sup_{\theta \in \Theta} F(\theta) - \hat{L}(\theta) = \sup_{\theta \in \Theta} F(\theta) - \frac{1}{n} \sum_{i=1}^n f(\langle w(\theta), x_i \rangle, y_i, \theta)$$

$$= \sup_{\theta \in \Theta, u \in \mathbb{R}^n} \inf_{\lambda \in \mathbb{R}^n} \langle \lambda, Xw(\theta) - u \rangle + F(\theta) - \frac{1}{n} \sum_{i=1}^n f(u_i, y_i, \theta).$$

Conditioned on $X w_1^* = \eta_1, ..., X w_k^* = \eta_k$ and $\xi$, the above is only random in $X$ by our multi-index model assumption on $y$. By Lemma 5, the above is equal in law to

$$\sup_{\theta \in \Theta, u \in \mathbb{R}^n} \inf_{\lambda \in \mathbb{R}^n} \left\langle \lambda, \left( \sum_{i=1}^k \eta_i (\Sigma w_i^*)^T + ZP\Sigma^{1/2} \right) w(\theta) - u \right\rangle + F(\theta) - \frac{1}{n} \sum_{i=1}^n f(u_i, y_i, \theta)$$

$$= \sup_{\theta \in \Theta, u \in \mathbb{R}^n} \inf_{\lambda \in \mathbb{R}^n} \left\langle \lambda, \left( ZP\Sigma^{1/2} \right) w(\theta) \right\rangle + \psi(u, \theta, \lambda \mid \eta_1, ..., \eta_k, \xi)$$

$$= \sup_{\substack{(w,u,\theta) \in \mathbb{R}^d \times \mathbb{R}^n \times \Theta \\ w = P\Sigma^{1/2} w(\theta)}} \inf_{\lambda \in \mathbb{R}^n} \langle \lambda, Zw \rangle + \psi(u, \theta, \lambda \mid \eta_1, ..., \eta_k, \xi)$$

$$= \Phi.$$

The function $\psi$ is continuous because we require $F$, $f$ and $w$ to be continuous in the definitions. $\quad\square$

Next, we are ready to apply Gaussian Minimax Theorem. Although the domains in (56) are not compact, we can use the truncation lemmas 1 and 2 in Appendix B.

**Lemma 7.** *In the same setting as Lemma 6, define the auxiliary problem as*

$$\Psi := \sup_{\substack{(u,\theta)\in\mathbb{R}^n\times\Theta \\ \langle H, P\Sigma^{1/2}w(\theta)\rangle \geq \left\|\|P\Sigma^{1/2}w(\theta)\|_2 G + \sum_{i=1}^k \langle w(\theta),\Sigma w_i^*\rangle\eta_i - u\right\|_2}} F(\theta) - \frac{1}{n}\sum_{i=1}^n f(u_i, y_i, \theta) \qquad (59)$$

*then for any $t \in \mathbb{R}$, it holds that*

$$\Pr\left(\sup_{\theta\in\mathcal{K}} F(\theta) - \hat{L}(\theta) > t\right) \leq 2\Pr(\Psi \geq t). \qquad (60)$$

*where the randomness in the second probability is taken over $G, H, \eta_1, ..., \eta_k$ and $\xi$.*

*Proof.* Denote $\mathcal{S}_r = \{(w, u, \theta) \in \mathbb{R}^d \times \mathbb{R}^n \times \Theta : w = P\Sigma^{1/2}w(\theta)$ and $\|w\|_2 + \|u\|_2 + \|\theta\|_2 \leq r\}$. The set $\mathcal{S}_r$ is bounded by definition and closed by the continuity of $w$. Hence, it is compact. Next, we denote the truncated problems:

$$\Phi_r := \sup_{(w,u,\theta)\in\mathcal{S}_r} \inf_{\lambda\in\mathbb{R}^n} \langle\lambda, Zw\rangle + \psi(u, \theta, \lambda \,|\, \eta_1, ..., \eta_k, \xi) \qquad (61)$$

$$\Phi_{r,s} := \sup_{(w,u,\theta)\in\mathcal{S}_r} \inf_{\|\lambda\|_2\leq s} \langle\lambda, Zw\rangle + \psi(u, \theta, \lambda \,|\, \eta_1, ..., \eta_k, \xi). \qquad (62)$$

By definition, we have $\Phi_r \leq \Phi_{r,s}$ and so

$$\Pr(\Phi_r > t) \leq \Pr(\Phi_{r,s} > t).$$

The corresponding auxiliary problems are

$$\Psi_{r,s} := \sup_{(w,u,\theta)\in\mathcal{S}_r} \inf_{\|\lambda\|_2\leq s} \|\lambda\|_2\langle H, w\rangle + \|w\|_2\langle G, \lambda\rangle + \psi(u, \theta, \lambda \,|\, \eta_1, ..., \eta_k, \xi)$$

$$= \sup_{(w,u,\theta)\in\mathcal{S}_r} \inf_{\|\lambda\|_2\leq s} \|\lambda\|_2\langle H, w\rangle + \left\langle\lambda, \|w\|_2 G + \sum_{i=1}^k \eta_i\langle w(\theta),\Sigma w_i^*\rangle - u\right\rangle$$

$$+ F(\theta) - \frac{1}{n}\sum_{i=1}^n f(u_i, g(\eta_{1,i}, ..., \eta_{k,i}, \xi_i), \theta)$$

$$= \sup_{(w,u,\theta)\in\mathcal{S}_r} \inf_{0\leq\lambda\leq s} \lambda\left(\langle H, w\rangle - \left\|\|w\|_2 G + \sum_{i=1}^k \eta_i\langle w(\theta),\Sigma w_i^*\rangle - u\right\|_2\right)$$

$$+ F(\theta) - \frac{1}{n}\sum_{i=1}^n f(u_i, g(\eta_{1,i}, ..., \eta_{k,i}, \xi_i), \theta)$$

and the limit of $s \to \infty$:

$$\Psi_r := \sup_{\substack{(w,u,\theta)\in\mathcal{S}_r \\ \langle H, w\rangle \geq \left\|\|w\|_2 G + \sum_{i=1}^k \eta_i\langle w(\theta),\Sigma w_i^*\rangle - u\right\|_2}} F(\theta) - \frac{1}{n}\sum_{i=1}^n f(u_i, g(\eta_{1,i}, ..., \eta_{k,i}, \xi_i), \theta)$$

By definition, it holds that $\Psi_r \leq \Psi$ and so

$$\Pr(\Psi_r \geq t) \leq \Pr(\Psi \geq t).$$

Thus, it holds that

$$\Pr(\Phi > t) = \lim_{r\to\infty} \Pr(\Phi_r > t) \qquad \text{by Lemma 1}$$

$$\leq \lim_{r\to\infty}\lim_{s\to\infty} \Pr(\Phi_{r,s} > t)$$

$$\leq 2\lim_{r\to\infty}\lim_{s\to\infty} \Pr(\Psi_{r,s} \geq t) \qquad \text{by Theorem 8}$$

$$= 2\lim_{r\to\infty} \Pr(\Psi_r \geq t) \qquad \text{by Lemma 2}$$

$$\leq 2\Pr(\Psi \geq t).$$

The proof concludes by applying Lemma 6 and the tower law. $\qquad \square$

The following two simple lemmas will be useful to analyze the auxiliary problem.

**Lemma 8.** *For $a, b, H > 0$, we have*

$$\sup_{\lambda \geq 0} -\lambda a + \frac{\lambda}{H + \lambda} b = (\sqrt{b} - \sqrt{Ha})_+^2.$$

*Proof.* Observe that

$$\sup_{\lambda \geq 0} -\lambda a + \frac{\lambda}{H + \lambda} b = b - \inf_{\lambda \geq 0} \lambda a + \frac{H}{H + \lambda} b.$$

Define $f(\lambda) = \lambda a + \frac{H}{H+\lambda} b$, then

$$f'(\lambda) = a - \frac{Hb}{(H + \lambda)^2} \leq 0 \iff (H + \lambda)^2 \leq \frac{Hb}{a}$$

$$\iff -\sqrt{\frac{Hb}{a}} - H \leq \lambda \leq \sqrt{\frac{Hb}{a}} - H$$

Since we require $\lambda \geq 0$, we only need to consider whether $\sqrt{\frac{Hb}{a}} - H \geq 0 \iff b \geq Ha$. If $b < Ha$, the infimum is attained at $\lambda = 0$. Otherwise, the infimum is attained at $\lambda^* = \sqrt{\frac{Hb}{a}} - H$, at which point

$$f(\lambda^*) = 2\sqrt{Hba} - Ha.$$

Plugging in, we see that the expression is equivalent to $(\sqrt{b} - \sqrt{Ha})_+^2$ in both cases. $\square$

**Lemma 9.** *For $a, b \geq 0$, we have*

$$\sup_{\lambda \geq 0} -\lambda a - \frac{b}{\lambda} = -\sqrt{4ab}$$

*Proof.* Define $f(\lambda) = -\lambda a - \frac{b}{\lambda}$, then

$$f'(\lambda) = -a + \frac{b}{\lambda^2} \geq 0 \iff \frac{b}{a} \geq \lambda^2$$

and so in the domain $\lambda \geq 0$, the optimum is attained at $\lambda^* = \sqrt{b/a}$ at which point $f(\lambda^*) = -2\sqrt{ab}$. $\square$

We are now ready to analyze the auxiliary problem.

**Lemma 10.** *In the same setting as in Lemma 6, assume that for every $\delta > 0$*

(A) $C_\delta : \mathbb{R}^d \to [0, \infty]$ *is a continuous function such that with probability at least $1 - \delta/4$ over $H \sim \mathcal{N}(0, I_d)$, uniformly over all $w \in \mathbb{R}^d$, we have that*

$$\langle \Sigma^{1/2} PH, w \rangle \leq C_\delta(w) \tag{63}$$

(B) $\epsilon_\delta$ *is a positive real number such that with probability at least $1 - \delta/4$ over $\{(\tilde{x}_i, \tilde{y}_i)\}_{i=1}^n$ drawn i.i.d. from $\tilde{D}$, it holds uniformly over all $\theta \in \Theta$ that*

$$\frac{1}{n} \sum_{i=1}^n f(\langle \phi(w(\theta)), \tilde{x}_i \rangle, \tilde{y}_i, \theta) \geq \frac{1}{1 + \epsilon_\delta} \mathbb{E}_{(\tilde{x}, \tilde{y}) \sim \tilde{D}}[f(\langle \phi(w(\theta)), \tilde{x} \rangle, \tilde{y}, \theta)]. \tag{64}$$

*where the distribution $\tilde{D}$ over $(\tilde{x}, \tilde{y})$ is given by*

$$\tilde{x} \sim \mathcal{N}(0, I_{k+1}), \quad \tilde{\xi} \sim \mathcal{D}_\xi, \quad \tilde{y} = g(\tilde{x}_1, ..., \tilde{x}_k, \tilde{\xi})$$

*and the mapping $\phi : \mathbb{R}^d \to \mathbb{R}^{k+1}$ is defined as*

$$\phi(w) = (\langle w, \Sigma w_1^* \rangle, ..., \langle w, \Sigma w_k^* \rangle, \|P\Sigma^{1/2} w\|_2)^T.$$

*Then the following is true:*

(i) *suppose for some choice of $M_\theta$ that is continuous in $\theta$, it holds for every $y \in \mathcal{Y}$ and $\theta \in \Theta$, $f$ is $M_\theta$-Lipschitz with respect to the first argument, then with probability at least $1 - \delta$, uniformly over all $\theta \in \Theta$, we have*

$$L(\theta) \leq (1 + \epsilon_\delta) \left( \hat{L}(\theta) + M_\theta \sqrt{\frac{C_\delta(w(\theta))^2}{n}} \right). \tag{65}$$

(ii) *suppose for some choice of $H_\theta$ that is continuous in $\theta$, it holds for every $y \in \mathcal{Y}$ and $\theta \in \Theta$, $f$ is non-negative and $\sqrt{f}$ is $\sqrt{H_\theta}$-Lipschitz with respect to the first argument, then with probability at least $1 - \delta$, uniformly over all $\theta \in \Theta$, we have*

$$L(\theta) \leq (1 + \epsilon_\delta) \left( \sqrt{\hat{L}(\theta)} + \sqrt{\frac{H_\theta C_\delta(w(\theta))^2}{n}} \right)^2. \tag{66}$$

*Proof.* First, let's simplify the auxiliary problem (59). Changing variables to subtract the quantity $G_i \left\| P\Sigma^{1/2} w(\theta) \right\|_2 + \sum_{l=1}^k \langle w(\theta), \Sigma w_l^* \rangle \eta_{l,i}$ from each of the former $u_i$, we have that

$$\Psi = \sup_{\substack{(u,\theta) \in \mathbb{R}^n \times \Theta \\ \|u\|_2 \leq \langle H, P\Sigma^{1/2} w(\theta) \rangle}} F(\theta) - \frac{1}{n} \sum_{i=1}^n f\left( u_i + G_i \left\| P\Sigma^{1/2} w(\theta) \right\|_2 + \sum_{l=1}^k \langle w(\theta), \Sigma w_l^* \rangle \eta_{l,i}, y_i, \theta \right)$$

and separating the optimization problem in $u$ and $\theta$, we obtain

$$\Psi = \sup_{\theta \in \Theta} F(\theta)$$

$$- \frac{1}{n} \inf_{\substack{u \in \mathbb{R}^n : \\ \|u\|_2 \leq \langle H, P\Sigma^{1/2} w(\theta) \rangle}} \sum_{i=1}^n f\left( u_i + G_i \left\| P\Sigma^{1/2} w(\theta) \right\|_2 + \sum_{l=1}^k \langle w(\theta), \Sigma w_l^* \rangle \eta_{l,i}, y_i, \theta \right).$$

Next, we will lower bound the infimum term by weak duality to obtain upper bound on $\Psi$:

$$\inf_{\substack{u \in \mathbb{R}^n : \\ \|u\|_2 \leq \langle H, P\Sigma^{1/2} w(\theta) \rangle}} \sum_{i=1}^n f\left( u_i + G_i \left\| P\Sigma^{1/2} w(\theta) \right\|_2 + \sum_{l=1}^k \langle w(\theta), \Sigma w_l^* \rangle \eta_{l,i}, y_i, \theta \right)$$

$$= \inf_{u \in \mathbb{R}^n} \sup_{\lambda \geq 0} \lambda (\|u\|_2^2 - \langle \Sigma^{1/2} PH, w(\theta) \rangle^2)$$

$$+ \sum_{i=1}^n f\left( u_i + G_i \left\| P\Sigma^{1/2} w(\theta) \right\|_2 + \sum_{l=1}^k \langle w(\theta), \Sigma w_l^* \rangle \eta_{l,i}, y_i, \theta \right)$$

$$\geq \sup_{\lambda \geq 0} -\lambda \langle \Sigma^{1/2} PH, w(\theta) \rangle^2$$

$$+ \inf_{u \in \mathbb{R}^n} \sum_{i=1}^n f\left( u_i + G_i \left\| P\Sigma^{1/2} w(\theta) \right\|_2 + \sum_{l=1}^k \langle w(\theta), \Sigma w_l^* \rangle \eta_{l,i}, y_i, \theta \right) + \lambda \|u\|_2^2$$

$$= \sup_{\lambda \geq 0} -\lambda \langle \Sigma^{1/2} PH, w(\theta) \rangle^2$$

$$+ \sum_{i=1}^n \inf_{u_i \in \mathbb{R}} f\left( u_i + G_i \left\| P\Sigma^{1/2} w(\theta) \right\|_2 + \sum_{l=1}^k \langle w(\theta), \Sigma w_l^* \rangle \eta_{l,i}, y_i, \theta \right) + \lambda u_i^2$$

$$= \sup_{\lambda \geq 0} -\lambda \langle \Sigma^{1/2} PH, w(\theta) \rangle^2 + \sum_{i=1}^n f_\lambda \left( G_i \left\| P\Sigma^{1/2} w(\theta) \right\|_2 + \sum_{l=1}^k \langle w(\theta), \Sigma w_l^* \rangle \eta_{l,i}, y_i, \theta \right).$$

Suppose that for every $y \in \mathcal{Y}$ and $\theta \in \Theta$, $f$ is $M_\theta$-Lipschitz with respect to the first argument, then by Proposition 2, the above can be further lower bounded by the following quantity:

$$\sup_{\lambda \geq 0} -\lambda \langle \Sigma^{1/2} PH, w(\theta) \rangle^2 - \frac{nM_\theta^2}{4\lambda} + \sum_{i=1}^n f\left( \sum_{l=1}^k \langle w(\theta), \Sigma w_l^* \rangle \eta_{l,i} + \left\| P\Sigma^{1/2} w(\theta) \right\|_2 G_i, y_i, \theta \right).$$

On the other hand, suppose that for every $y \in \mathcal{Y}$ and $\theta \in \Theta$, $f$ is non-negative and $\sqrt{f}$ is $\sqrt{H_\theta}$-Lipschitz with respect to the first argument, then by Proposition 1, the above can be further lower bounded by:

$$\sup_{\lambda \geq 0} -\lambda \langle \Sigma^{1/2} PH, w(\theta) \rangle^2 + \frac{\lambda}{H_\theta + \lambda} \left[ \sum_{i=1}^{n} f\left( \sum_{l=1}^{k} \langle w(\theta), \Sigma w_l^* \rangle \eta_{l,i} + \left\| P\Sigma^{1/2} w(\theta) \right\|_2 G_i, y_i, \theta \right) \right].$$

Notice that if we write $\tilde{x}_i = (\eta_{1,i}, ..., \eta_{k,i}, G_i)$, then $(\tilde{x}_i, y_i)$ are independent with distribution exactly equal to $\tilde{\mathcal{D}}$. Moreover, we have

$$f\left( \sum_{l=1}^{k} \langle w(\theta), \Sigma w_l^* \rangle \eta_{l,i} + \left\| P\Sigma^{1/2} w(\theta) \right\|_2 G_i, y_i, \theta \right) = f(\langle \phi(w(\theta)), \tilde{x}_i \rangle, y_i, \theta)$$

and it is easy to see that the joint distribution of $(\langle \phi(w(\theta)), \tilde{x} \rangle, y)$ with $(\tilde{x}, y) \sim \tilde{\mathcal{D}}$ is exactly the same as $(\langle w(\theta), x \rangle, y)$ with $(x, y) \sim \mathcal{D}$. As a result, we have that

$$\mathbb{E}_{(\tilde{x},y) \sim \tilde{D}}[f(\langle \phi(w(\theta)), \tilde{x} \rangle, y, \theta)] = L(\theta).$$

By our assumption (63), (64) and a union bound, we have with probability at least $1 - \delta/2$

$$|\langle \Sigma^{1/2} PH, w(\theta) \rangle| \leq C_\delta(w(\theta))$$

$$\frac{1}{n} \sum_{i=1}^{n} f\left( \sum_{l=1}^{k} \langle w(\theta), \Sigma w_l^* \rangle \eta_{l,i} + \left\| P\Sigma^{1/2} w(\theta) \right\|_2 G_i, y_i, \theta \right) \geq \frac{1}{1 + \epsilon_\delta} L(\theta).$$

Therefore, if $f$ is $M_\theta$-Lipschitz, then by by Lemma 9, we have

$$\Psi \leq \sup_{\theta \in \Theta} F(\theta) - \sup_{\lambda \geq 0} -\lambda \frac{C_\delta(w(\theta))^2}{n} - \frac{M_\theta^2}{4\lambda} + \frac{1}{1 + \epsilon_\delta} L(\theta)$$

$$= \sup_{\theta \in \Theta} F(\theta) + \sqrt{M_\theta^2 \frac{C_\delta(w(\theta))^2}{n}} - \frac{1}{1 + \epsilon_\delta} L(\theta)$$

Consequently, by taking $F(\theta) = \frac{1}{1+\epsilon_\delta} L(\theta) - M_\theta \sqrt{\frac{C_\delta(w(\theta))^2}{n}}$ and Lemma 7, we have shown that with probability at least $1 - \delta$, we have

$$\sup_{\theta \in \mathcal{K}} F(\theta) - \hat{L}(\theta) \leq 0 \implies \frac{1}{1 + \epsilon_\delta} L(\theta) \leq \hat{L}(\theta) + M_\theta \sqrt{\frac{C_\delta(w(\theta))^2}{n}}.$$

If $\sqrt{f}$ is $\sqrt{H_\theta}$-Lipschitz, then by Lemma 8

$$\Psi \leq \sup_{\theta \in \mathcal{K}} F(\theta) - \sup_{\lambda \geq 0} -\lambda \frac{C_\delta(w(\theta))^2}{n} + \frac{\lambda}{H_\theta + \lambda} \frac{1}{1 + \epsilon_\delta} L(\theta)$$

$$= \sup_{\theta \in \mathcal{K}} F(\theta) - \left( \sqrt{\frac{L(\theta)}{1 + \epsilon_\delta}} - \sqrt{\frac{H_\theta C_\delta(w(\theta))^2}{n}} \right)_+^2.$$

Consequently, by taking $F(\theta) = \left( \sqrt{\frac{L(\theta)}{1+\epsilon_\delta}} - \sqrt{\frac{H_\theta C_\delta(w(\theta))^2}{n}} \right)_+^2$ and Lemma 7, we have shown that with probability at least $1 - \delta$, we have

$$\sup_{\theta \in \mathcal{K}} F(\theta) - \hat{L}(\theta) \leq 0.$$

Rearranging, either we have

$$\sqrt{\frac{L(\theta)}{1 + \epsilon_\delta}} - \sqrt{\frac{H_\theta C_\delta(w(\theta))^2}{n}} < 0 \implies L(\theta) < (1 + \epsilon_\delta) \frac{H_\theta C_\delta(w(\theta))^2}{n}$$

or we have

$$\sqrt{\frac{L(\theta)}{1+\epsilon_\delta}} - \sqrt{\frac{H_\theta C_\delta(w(\theta))^2}{n}} \geq 0 \implies \left(\sqrt{\frac{L(\theta)}{1+\epsilon_\delta}} - \sqrt{\frac{H_\theta C_\delta(w(\theta))^2}{n}}\right)^2 \leq \hat{L}(\theta)$$

$$\implies L(\theta) \leq (1+\epsilon_\delta)\left(\sqrt{\hat{L}(\theta)} + \sqrt{\frac{H_\theta C_\delta(w(\theta))^2}{n}}\right)^2.$$

In either case, the desired bound holds. $\qquad\square$

Finally, we are ready to prove Theorem 6. In the version below, we also provide uniform convergence guarantee (with sharp constant) for Lipschitz loss.

**Theorem 14.** *Suppose that assumptions (A), (B), (E) and (F) hold. For any $\delta \in (0,1)$, let $C_\delta : \mathbb{R}^d \to [0,\infty]$ be a continuous function such that with probability at least $1 - \delta/4$ over $x \sim \mathcal{N}(0,\Sigma)$, uniformly over all $\theta \in \Theta$,*

$$\langle w(\theta), Q^T x \rangle \leq C_\delta(w(\theta)). \tag{67}$$

*Then it holds that*

(i) *if for each $\theta \in \Theta$ and $y \in \mathcal{Y}$, $f$ is $M_\theta$-Lipschitz with respect to the first argument and $M_\theta$ is continuous in $\theta$, then with probability at least $1 - \delta$, it holds that uniformly over all $\theta \in \Theta$, we have*

$$(1-\epsilon) L(\theta) \leq \hat{L}(\theta) + M_\theta \sqrt{\frac{C_\delta(w(\theta))^2}{n}} \tag{68}$$

(ii) *if for each $\theta \in \Theta$ and $y \in \mathcal{Y}$, $f$ is non-negative and $\sqrt{f}$ is $\sqrt{H_\theta}$-Lipschitz with respect to the first argument, and $H_\theta$ is continuous in $\theta$, then with probability at least $1 - \delta$, it holds that uniformly over all $\theta \in \Theta$, we have*

$$(1-\epsilon) L(\theta) \leq \left(\sqrt{\hat{L}(\theta)} + \sqrt{\frac{H_\theta C_\delta(w(\theta))^2}{n}}\right)^2 \tag{69}$$

*where $\epsilon = O\left(\tau \sqrt{\frac{h\log(n/h) + \log(1/\delta)}{n}}\right)$.*

*Proof.* We apply the reduction argument at the beginning of the appendix. Given $\mathcal{D}$ that satisfies assumptions (A) and (B), we define $[\tilde{w}_1^*, ..., \tilde{w}_k^*] = \tilde{W} = W(W^T\Sigma W)^{-1/2}$ and $\tilde{f}, \tilde{g}$ as in (54) and (55). For $\{(x_i, y_i)\}_{i=1}^n$ sampled independently from $\mathcal{D}$, we observe that the joint distribution of $(x_i - \mu, y_i)$ can also be described by $\mathcal{D}'$ as follows:

(A') $x \sim \mathcal{N}(0, \Sigma)$

(B') $y = \tilde{g}(\eta_1, ..., \eta_k, \xi)$ where $\eta_i = \langle x, \tilde{w}_i \rangle$.

Indeed, we can check that

$$y = g(x^T W, \xi)$$
$$= g((x-\mu)^T\tilde{W}(W^T\Sigma W)^{1/2} + \mu^T W, \xi)$$
$$= \tilde{g}((x-\mu)^T\tilde{W}, \xi).$$

Moreover, by construction, we have

$$\hat{L}(\theta) = \frac{1}{n}\sum_{i=1}^n \tilde{f}(\langle w(\theta), x_i - \mu \rangle, y_i, \theta)$$

$$L(\theta) = \mathbb{E}_{\mathcal{D}'}\tilde{f}(\langle w(\theta), x_i \rangle, y_i, \theta)$$

and $\mathcal{D}'$ satisfies assumptions (A) and (B) with $\mu = 0$ and orthonormal $\Sigma^{1/2}\tilde{w}_1^*, ..., \Sigma^{1/2}\tilde{w}_1^*$ and falls into the setting in Lemma 6. We see that $f$ being Lipschitz or square-root Lipschitz is equivalent to

$\tilde{f}$ being Lipschitz or square-root Lipschitz. It remains to check assumptions (63) and (64) and then apply Lemma 10. Observe that

$$
\begin{aligned}
\Sigma^{-1/2}P\Sigma^{1/2} &= \Sigma^{-1/2}\left(I_d - \Sigma^{1/2}\tilde{W}\tilde{W}^T\Sigma^{1/2}\right)\Sigma^{1/2} \\
&= I_d - \tilde{W}\tilde{W}^T\Sigma = I - W(W^T\Sigma W)^{-1}W^T\Sigma \\
&= Q
\end{aligned}
\tag{70}
$$

and so $\Sigma^{1/2}P = Q^T\Sigma^{1/2}$.

To check that (63) holds, observe that $\langle\Sigma^{1/2}PH, w\rangle$ has the same distribution as $\langle Qw, x\rangle$. To check that (64) holds, we will apply Theorem 13. Note that the joint distribution of $(\langle\phi(w(\theta)), \tilde{x}\rangle, \tilde{y})$ with $(\tilde{x}, \tilde{y}) \sim \tilde{\mathcal{D}}$ is exactly the same as $(\langle w(\theta), x\rangle, y)$ with $(x, y) \sim \mathcal{D}'$ and so

$$
\frac{\mathbb{E}_{\tilde{\mathcal{D}}}[\tilde{f}(\langle\phi(w(\theta)), x\rangle, y, \theta)^4]^{1/4}}{\mathbb{E}_{\tilde{\mathcal{D}}}[\tilde{f}(\langle\phi(w(\theta)), x\rangle, y, \theta)]} = \frac{\mathbb{E}_{\mathcal{D}'}[\tilde{f}(\langle w(\theta), x\rangle, y, \theta)^4]^{1/4}}{\mathbb{E}_{\mathcal{D}'}[\tilde{f}(\langle w(\theta), x\rangle, y, \theta)]} = \frac{\mathbb{E}_{\mathcal{D}}[f(\langle w(\theta), x\rangle, y, \theta)^4]^{1/4}}{\mathbb{E}_{\mathcal{D}}[f(\langle w(\theta), x\rangle, y, \theta)]}.
$$

Therefore, the assumption (E) is equivalent to the hypercontractivity condition in Theorem 13. Note that $\{(x, y) \mapsto \mathbb{1}\{\tilde{f}(\langle\phi(w(\theta)), x\rangle, y, \theta) > t\} : (\theta, t) \in \Theta \times \mathbb{R}\}$ is a subclass of $\{(x, y) \mapsto \mathbb{1}\{f(\langle w, x\rangle + b, y, \theta) > t\} : (w, b, t, \theta) \in \mathbb{R}^{k+1} \times \mathbb{R} \times \mathbb{R} \times \Theta\}$. Therefore, by assumption (F), we can apply Theorem 13 and (64) holds. $\qquad\square$

## D  Norm Bounds

The following lemma is a version of Lemma 7 of Koehler et al. (2021) and follows straightforwardly from CGMT (Theorem 7), though it requires a slightly different truncation argument compared to the proof Theorem 6. For simplicity, we won't repeat the proof here and simply use it for our applications.

**Lemma 11** (Koehler et al. 2021, Lemma 7). *Let $Z : n \times d$ be a matrix with i.i.d. $\mathcal{N}(0, 1)$ entries and suppose $G \sim \mathcal{N}(0, I_n)$ and $H \sim \mathcal{N}(0, I_d)$ are independent of $Z$ and each other. Fix an arbitrary norm $\|\cdot\|$, any covariance matrix $\Sigma$, and any non-random vector $\xi \in \mathbb{R}^n$, consider the Primary Optimization (PO) problem:*

$$
\Phi := \min_{\substack{w \in \mathbb{R}^d : \\ Z\Sigma^{1/2}w = \xi}} \|w\|
\tag{71}
$$

*and the Auxiliary Optimization (AO) problem:*

$$
\Psi := \min_{\substack{w \in \mathbb{R}^d : \\ \|G\|\|\Sigma^{1/2}w\|_2 - \xi\|_2 \leq \langle\Sigma^{1/2}H, w\rangle}} \|w\|.
\tag{72}
$$

*Then for any $t \in \mathbb{R}$, it holds that*

$$
\Pr(\Phi > t) \leq 2\Pr(\phi \geq t).
\tag{73}
$$

The next lemma analyzes the AO in Lemma 11. Our proof closely follows Lemma 8 of Koehler et al. 2021, but we don't make assumptions on $\xi$ yet to allow more applications.

**Lemma 12.** *Let $Z : n \times d$ be a matrix with i.i.d. $\mathcal{N}(0, 1)$ entries. Fix any $\delta > 0$, covariance matrix $\Sigma$ and non-random vector $\xi \in \mathbb{R}^n$, then there exists $\epsilon \lesssim \log(1/\delta)\left(\frac{1}{n} + \frac{1}{\sqrt{R(\Sigma)}} + \frac{n}{R(\Sigma)}\right)$ such that with probability at least $1 - \delta$, it holds that*

$$
\min_{\substack{w \in \mathbb{R}^d : \\ Z\Sigma^{1/2}w = \xi}} \|w\|_2^2 \leq (1 + \epsilon)\frac{\|\xi\|_2^2}{\mathrm{Tr}(\Sigma)}.
\tag{74}
$$

*Proof.* By a union bound, there exists a constant $C > 0$ such that the following events occur together with probability at least $1 - \delta/2$:

1. Since $\langle G, \xi \rangle \sim \mathcal{N}(0, \|\xi\|_2^2)$, by the standard Gaussian tail bound $\Pr(|Z| \geq t) \leq 2e^{-t^2/2}$, we have

$$|\langle G, \xi \rangle| \leq \|\xi\|_2 \sqrt{2 \log(32/\delta)}$$

2. Using subexponential Bernstein's inequality (Theorem 2.8.1 of Vershynin (2018)), requiring $n = \Omega(\log(1/\delta))$, we have

$$\|G\|_2^2 \leq 2n$$

3. Using the first part of Lemma 4, we have

$$\|\Sigma^{1/2} H\|_2^2 \geq \operatorname{Tr}(\Sigma) \left( 1 - C \frac{\log(32/\delta)}{\sqrt{R(\Sigma)}} \right)$$

4. Using the last part of Lemma 4, requiring $R(\Sigma) \gtrsim \log(32/\delta)^2$

$$\frac{\|\Sigma H\|_2^2}{\|\Sigma^{1/2} H\|_2^2} \leq C \log(32/\delta) \frac{\operatorname{Tr}(\Sigma^2)}{\operatorname{Tr}(\Sigma)}$$

Therefore, by the AM-GM inequality, it holds that

$$\begin{aligned}
\|G\| \Sigma^{1/2} w\|_2 - \xi\|_2^2 &= \|G\|_2^2 \|\Sigma^{1/2} w\|_2^2 + \|\xi\|_2^2 - 2\langle G, \xi \rangle \|\Sigma^{1/2} w\|_2 \\
&\leq 2n \|\Sigma^{1/2} w\|_2^2 + \|\xi\|_2^2 + 2\|\xi\|_2 \sqrt{2 \log(32/\delta)} \|\Sigma^{1/2} w\|_2 \\
&\leq 3n \|\Sigma^{1/2} w\|_2^2 + \left( 1 + \frac{2 \log(32/\delta)}{n} \right) \|\xi\|_2^2.
\end{aligned}$$

To apply lemma 11, we will consider $w$ of the form $w = \alpha \frac{\Sigma^{1/2} H}{\|\Sigma^{1/2} H\|_2}$ for some $\alpha > 0$. Then we have

$$\|G\| \Sigma^{1/2} w\|_2 - \xi\|_2^2 \leq 3nC \log(32/\delta) \frac{\operatorname{Tr}(\Sigma^2)}{\operatorname{Tr}(\Sigma)} \alpha^2 + \left( 1 + \frac{2 \log(32/\delta)}{n} \right) \|\xi\|_2^2$$

and

$$\langle \Sigma^{1/2} H, w \rangle^2 = \alpha^2 \|\Sigma^{1/2} H\|_2^2 \geq \alpha^2 \operatorname{Tr}(\Sigma) \left( 1 - C \frac{\log(32/\delta)}{\sqrt{R(\Sigma)}} \right).$$

So it suffices to choose $\alpha$ such that

$$\begin{aligned}
\alpha^2 &\geq \frac{\left( 1 + \frac{2 \log(32/\delta)}{n} \right) \|\xi\|_2^2}{\operatorname{Tr}(\Sigma) \left( 1 - C \frac{\log(32/\delta)}{\sqrt{R(\Sigma)}} \right) - 3nC \log(32/\delta) \frac{\operatorname{Tr}(\Sigma^2)}{\operatorname{Tr}(\Sigma)}} \\
&= \frac{1 + \frac{2 \log(32/\delta)}{n}}{1 - C \log(32/\delta) \left( \frac{1}{\sqrt{R(\Sigma)}} + 3 \frac{n}{R(\Sigma)} \right)} \frac{\|\xi\|_2^2}{\operatorname{Tr}(\Sigma)}
\end{aligned}$$

and we are done. $\qquad \square$

A challenge for analyzing the minimal norm to interpolate is that the projection matrix $Q$ is not necessarily an orthogonal projection. However, the following lemma suggests that if $\Sigma^\perp = Q^T \Sigma Q$ has high effective rank, then we can let $R$ be the orthogonal projection matrix onto the image of $Q$ and $R \Sigma R$ is approximately the same as $\Sigma^\perp$ in terms of the quantities that are relevant to the norm analysis.

**Lemma 13.** *Consider $Q = I - \sum_{i=1}^{k} w_i^* (w_i^*)^T \Sigma$ where $\Sigma^{1/2} w_1^*, ..., \Sigma^{1/2} w_k^*$ are orthonormal and we let $R$ be the orthogonal projection matrix onto the image of $Q$. Then it holds that $\operatorname{rank}(R) = d - k$ and*

$$R \Sigma w_i^* = 0 \quad \text{for any } i = 1, ..., k.$$

*Moreover, we have $QR = R$ and $RQ = Q$, and so*

$$\frac{1}{\operatorname{Tr}(R \Sigma R)} \leq \left( 1 - \frac{k}{n} - \frac{n}{R(Q^T \Sigma Q)} \right)^{-1} \frac{1}{\operatorname{Tr}(Q^T \Sigma Q)}$$

$$\frac{n}{R(R \Sigma R)} \leq \left( 1 - \frac{k}{n} - \frac{n}{R(Q^T \Sigma Q)} \right)^{-2} \frac{n}{R(Q^T \Sigma Q)}.$$

*Proof.* It is obvious that $\mathrm{rank}(R) = \mathrm{rank}(Q)$ and by the rank-nullity theorem, it suffices to show the nullity of $Q$ is $k$. To this end, we observe that

$$Qw = 0 \iff \Sigma^{-1/2}\left(I - \sum_{i=1}^{k}(\Sigma^{1/2}w_i^*)(\Sigma^{1/2}w_i^*)^T\right)\Sigma^{1/2}w = 0$$

$$\iff \left(I - \sum_{i=1}^{k}(\Sigma^{1/2}w_i^*)(\Sigma^{1/2}w_i^*)^T\right)\Sigma^{1/2}w = 0$$

$$\iff \Sigma^{1/2}w \in \mathrm{span}\{\Sigma^{1/2}w_1^*, ..., \Sigma^{1/2}w_k^*\}$$

$$\iff w \in \mathrm{span}\{w_1^*, ..., w_k^*\}.$$

It is also straightforward to verify that $Q^2 = Q$ and $Q^T\Sigma w_i^* = 0$ for $i = 1, ..., k$. For any $v \in \mathbb{R}^d$, $Rv$ lies in the image of $Q$ and so there exists $w$ such that $Rv = Qw$. Then we can check that

$$v^T R\Sigma w_i^* = \langle Rv, \Sigma w_i^*\rangle$$
$$= \langle Qw, \Sigma w_i^*\rangle = \langle w, Q^T\Sigma w_i^*\rangle = 0$$

and

$$(QR)v = Q(Rv)$$
$$= Q(Qw) = Q^2 w$$
$$= Qw = Rv.$$

Since the choice of $v$ is arbitrary, it must be the case that $R\Sigma w_i^* = 0$ and $QR = R$. For any $v \in \mathbb{R}^d$, we can check

$$(RQ)v = R(Qv) = Qv$$

by the definition of orthogonal projection. Therefore, it must be the case that $RQ = Q$. Finally, we use $R = QR = RQ^T$ to show that

$$\mathrm{Tr}(R\Sigma R) = \mathrm{Tr}(RQ^T\Sigma QR) = \mathrm{Tr}(Q^T\Sigma QR)$$
$$= \mathrm{Tr}(Q^T\Sigma Q) - \mathrm{Tr}(Q^T\Sigma Q(I - R))$$
$$\geq \mathrm{Tr}(Q^T\Sigma Q) - \sqrt{\mathrm{Tr}((Q^T\Sigma Q)^2)\,\mathrm{Tr}((I - R)^2)}$$
$$= \mathrm{Tr}(Q^T\Sigma Q)\left(1 - \sqrt{\frac{k}{R(Q^T\Sigma Q)}}\right)$$
$$= \mathrm{Tr}(Q^T\Sigma Q)\left(1 - \frac{k}{n} - \frac{n}{R(Q^T\Sigma Q)}\right)$$

and

$$\mathrm{Tr}((R\Sigma R)^2) = \mathrm{Tr}(\Sigma R\Sigma R)$$
$$= \mathrm{Tr}(\Sigma QRQ^T\Sigma QRQ^T)$$
$$= \mathrm{Tr}((RQ^T\Sigma Q)R(Q^T\Sigma QR))$$
$$\leq \mathrm{Tr}((RQ^T\Sigma Q)(Q^T\Sigma QR)) = \mathrm{Tr}((Q^T\Sigma Q)^2 R)$$
$$\leq \mathrm{Tr}((Q^T\Sigma Q)^2).$$

Rearranging concludes the proof. $\qquad\square$

## D.1 Phase Retrieval

**Theorem 2.** *Under assumptions (A) and (B), let $f : \mathbb{R} \times \mathcal{Y} \to \mathbb{R}$ be given by $f(\hat{y}, y) := (|\hat{y}| - y)^2$ with $\mathcal{Y} = \mathbb{R}_{\geq 0}$. Let $Q$ be the same as in Theorem 1 and $\Sigma^\perp = Q^T\Sigma Q$. Fix any $w^\sharp \in \mathbb{R}^d$ such that $Qw^\sharp = 0$ and for some $\rho \in (0, 1)$, it holds that*

$$\hat{L}_f(w^\sharp) \leq (1 + \rho)L_f(w^\sharp). \tag{9}$$

*Then with probability at least $1 - \delta$, for some $\epsilon \lesssim \rho + \log\left(\frac{1}{\delta}\right)\left(\frac{1}{\sqrt{n}} + \frac{1}{\sqrt{R(\Sigma^\perp)}} + \frac{k}{n} + \frac{n}{R(\Sigma^\perp)}\right)$, it holds that*

$$\min_{\substack{w \in \mathbb{R}^d: \\ \forall i \in [n], \langle w, x_i\rangle^2 = y_i^2}} \|w\|_2 \leq \|w^\sharp\|_2 + (1 + \epsilon)\sqrt{\frac{nL_f(w^\sharp)}{\text{Tr}(\Sigma^\perp)}}. \tag{10}$$

*Proof.* Without loss of generality, we assume that $\mu$ lies in the span of $\{\Sigma w_1^*, ..., \Sigma w_k^*\}$ because otherwise we can simply increase $k$ by one. Moreover, we can assume that $\{\Sigma^{1/2} w_1^*, ..., \Sigma^{1/2} w_k^*\}$ are orthonormal because otherwise we let $\tilde{W} = W(W^T \Sigma W)^{-1}$ and conditioning on $W^T(x - \mu)$ is the same as conditioning on $\tilde{W}^T(x - \mu)$. By Lemma 5, conditioned on

$$\begin{pmatrix} \eta_1^T \\ ... \\ \eta_k^T \end{pmatrix} = [W^T(x_1 - \mu), ..., W^T(x_n - \mu)]$$

the distribution of $X$ is the same as

$$X = 1\mu^T + \sum_{i=1}^{k} \eta_i (\Sigma w_i^*)^T + Z\Sigma^{1/2}Q$$

where $Z$ has i.i.d. standard normal entries. Furthermore, conditioned on $W^T(x - \mu)$ and the noise of variable in $y$ (which is independent of $x$), by the multi-index assumption (B), the label $y$ is non-random. Since $Qw^\sharp = 0$, we have $w^\sharp = \sum_{i=1}^{k} \langle w_i^*, \Sigma w^\sharp \rangle w_i^*$ and so

$$\langle w^\sharp, x \rangle = \langle w^\sharp, \mu \rangle + \sum_{i=1}^{k} \langle w_i^*, \Sigma w^\sharp \rangle \langle w_i^*, x - \mu \rangle.$$

Therefore, $\langle w^\sharp, x \rangle$ also becomes non-random after conditioning. We can let $I = \{i \in [n] : \langle w^\sharp, x_i \rangle \geq 0\}$ and define $\xi \in \mathbb{R}^n$ by

$$\xi_i = \begin{cases} y_i - |\langle w^\sharp, x_i \rangle| & \text{if } i \in I \\ |\langle w^\sharp, x_i \rangle| - y_i & \text{if } i \notin I \end{cases}$$

and $\xi$ is non-random after conditioning. Following the construction discussed in the main text, for any $w^\sharp \in \mathbb{R}^d$, the predictor $w = w^\sharp + w^\perp$ satisfies $|\langle w, x_i \rangle| = y_i$ where

$$w^\perp = \arg\min_{\substack{w \in \mathbb{R}^d: \\ Xw = \xi}} \|w\|_2$$

by the definition of $\xi$. Hence, we have

$$\min_{w \in \mathbb{R}^d: \forall i \in [n], \langle w, x_i\rangle^2 = y_i^2} \|w\|_2 \leq \|w^\sharp\|_2 + \|w^\perp\|_2$$

and it suffices to control $\|w^\perp\|_2$.

Let $R$ be the orthogonal projection matrix onto the image of $Q$ and we consider $w$ of the form $Rw$ to upper bound $\|w^\perp\|_2$. By Lemma 13, we know $QR = R$ and $R\Sigma w_i^* = 0$. By the assumption that $\mu$ lies in the span of $\{\Sigma w_1^*, ..., \Sigma w_k^*\}$, we have

$$\left(1\mu^T + \sum_{i=1}^{k} \eta_i (\Sigma w_i^*)^T + Z\Sigma^{1/2}Q\right) Rw = Z\Sigma^{1/2}Rw.$$

Since $R$ is an orthogonal projection, it holds that $\|Rw\|_2 \leq \|w\|_2$. Finally, we observe that the distribution of $Z\Sigma^{1/2}R$ is the same as $Z(R\Sigma R)^{1/2}$ and so

$$\|w^\perp\|_2 \leq \min_{\substack{w \in \mathbb{R}^d: \\ Z(R\Sigma R)^{1/2}w = \xi}} \|w\|_2.$$

We are now ready to apply Lemma 12 to the covariance $R\Sigma R$. We are allowed to replace the dependence on $R\Sigma R$ by the dependence on $\Sigma^\perp$ by the last two inequalities of Lemma 13. The desired conclusion follows by the observation that $\|\xi\|_2^2 = n\hat{L}_f(w^\sharp)$ and the assumption that $\hat{L}_f(w^\sharp) \leq (1 + \rho)L_f(w^\sharp)$. $\qquad\square$

## D.2   ReLU Regression

The proof of Theorem 3 will closely follow the proof of Theorem 2.

**Theorem 3.** *Under assumptions (A) and (B), let $f : \mathbb{R} \times \mathcal{Y} \to \mathbb{R}$ be the loss defined in (13) with $\mathcal{Y} = \mathbb{R}_{\geq 0}$. Let $Q$ be the same as in Theorem 1 and $\Sigma^\perp = Q^T \Sigma Q$. Fix any $(w^\sharp, b^\sharp) \in \mathbb{R}^{d+1}$ such that $Qw^\sharp = 0$ and for some $\rho \in (0, 1)$, it holds that*

$$\hat{L}_f(w^\sharp, b^\sharp) \leq (1 + \rho) L_f(w^\sharp, b^\sharp). \tag{14}$$

*Then with probability at least $1 - \delta$, for some $\epsilon \lesssim \rho + \log\left(\frac{1}{\delta}\right)\left(\frac{1}{\sqrt{n}} + \frac{1}{\sqrt{R(\Sigma^\perp)}} + \frac{k}{n} + \frac{n}{R(\Sigma^\perp)}\right)$, it holds that*

$$\min_{\substack{(w,b)\in\mathbb{R}^{d+1}: \\ \forall i\in[n], \sigma(\langle w, x_i\rangle + b) = y_i}} \|w\|_2 \leq \|w^\sharp\|_2 + (1 + \epsilon)\sqrt{\frac{n L_f(w^\sharp, b^\sharp)}{\mathrm{Tr}(\Sigma^\perp)}}. \tag{15}$$

*Proof.* We let $I = \{i \in [n] : y_i > 0\}$ and for any $(w^\sharp, b^\sharp) \in \mathbb{R}^{d+1}$, we define $\xi \in \mathbb{R}^n$ by

$$\xi_i = \begin{cases} y_i - \langle w^\sharp, x_i\rangle - b^\sharp & \text{if } i \in I \\ -\sigma(\langle w^\sharp, x_i\rangle + b^\sharp) & \text{if } i \notin I. \end{cases}$$

By the definition of $\xi$, the predictor $(w, b) = (w^\sharp + w^\perp, b^\sharp)$ satisfies $\sigma(\langle w, x_i\rangle + b) = y_i$ where

$$w^\perp = \arg\min_{\substack{w\in\mathbb{R}^d: \\ Xw=\xi}} \|w\|_2.$$

Hence, we have

$$\min_{\substack{(w,b)\in\mathbb{R}^{d+1}: \\ \forall i\in[n], \sigma(\langle w, x_i\rangle + b) = y_i}} \|w\|_2 \leq \|w^\sharp\|_2 + \|w^\perp\|_2$$

and it suffices to control $\|w^\perp\|_2$.

Similar to the proof of Theorem 2, we make the simplifying assumption that $\mu$ lies in the span of $\{\Sigma w_1^*, ..., \Sigma w_k^*\}$ and $\{\Sigma^{1/2} w_1^*, ..., \Sigma^{1/2} w_k^*\}$ are orthonormal. Conditioned on $W^T(x_i - \mu)$ and the noise variable in $y_i$, both $y_i$ and $\langle w^\sharp, x_i\rangle$ are non-random, and so $\xi$ is also non-random. The distribution of $X$ is the same as

$$X = 1\mu^T + \sum_{i=1}^{k} \eta_i (\Sigma w_i^*)^T + Z\Sigma^{1/2} Q.$$

If we consider $w$ of the form $Rw$, then we have

$$\|w^\perp\|_2 \leq \min_{\substack{w\in\mathbb{R}^d: \\ Z(R\Sigma R)^{1/2}w=\xi}} \|w\|_2.$$

We are now ready to apply Lemma 12 to the covariance $R\Sigma R$. We are allowed to replace the dependence on $R\Sigma R$ by the dependence on $\Sigma^\perp$ by the last two inequalities of Lemma 13. The desired conclusion follows by the observation that $\|\xi\|_2^2 = n\hat{L}_f(w^\sharp, b^\sharp)$ due to the definition (13) and the assumption that $\hat{L}_f(w^\sharp) \leq (1 + \rho) L_f(w^\sharp, b^\sharp)$. $\qquad\square$

## D.3   Low-rank Matrix Sensing

**Theorem 4.** *Suppose that $d_1 d_2 > n$, then there exists some $\epsilon \lesssim \sqrt{\frac{\log(32/\delta)}{n}} + \frac{n}{d_1 d_2}$ such that with probability at least $1 - \delta$, it holds that*

$$\min_{\forall i\in[n], \langle A_i, X\rangle = y_i} \|X\|_* \leq \sqrt{r}\|X^*\|_F + (1 + \epsilon)\sqrt{\frac{n\sigma^2}{d_1 \vee d_2}}. \tag{17}$$

*Proof.* Without loss of generality, we will assume that $d_1 \leq d_2$. We will vectorize the measurement matrices and estimator $A_1, ..., A_n, X \in \mathbb{R}^{d_1 \times d_2}$ as $a_1, ..., a_n, x \in \mathbb{R}^{d_1 d_2}$ and define $\|x\|_* = \|X\|_*$. Denote $A = [a_1, ..., a_n]^T \in \mathbb{R}^{n \times d_1 d_2}$. We define the primary problem $\Phi$ by

$$\Phi := \min_{\forall i \in [n], \langle A_i, X \rangle = \xi} \|X\|_* = \min_{Ax = \xi} \|x\|_*.$$

By Lemma 11, it suffices to consider the auxiliary problem

$$\Psi := \min_{\|G\|x\|_2 - \xi\|_2 \leq -\langle H, x \rangle} \|x\|_*.$$

We will pick $x$ of the form $x = -\alpha H$ for some $\alpha \geq 0$, which needs to satisfy $\alpha \|H\|_2^2 \geq \|\alpha G\|H\|_2 - \xi\|_2$. By a union bound, the following events occur simultaneously with probability at least $1 - \delta/2$:

1. by Lemma 3, it holds that

$$\|G\|_2 \leq \sqrt{n} + 2\sqrt{\log(32/\delta)}$$

$$\frac{\|\xi\|_2}{\sigma} \leq \sqrt{n} + 2\sqrt{\log(32/\delta)}$$

$$\|H\|_2 \leq \sqrt{d_1 d_2} + 2\sqrt{\log(32/\delta)}$$

2. Condition on $\xi$, we have $\frac{1}{\|\xi\|}\langle G, \xi \rangle \sim \mathcal{N}(0, 1)$ and so by standard Gaussian tail bound $\Pr(|Z| > t) \leq 2e^{-t^2/2}$

$$\frac{|\langle G, \xi \rangle|}{\|\xi\|} \leq \sqrt{2\log(16/\delta)}$$

Then we can use AM-GM inequality to show for sufficiently large $n$

$$\|\alpha G\|H\|_2 - \xi\|_2^2$$
$$= \alpha^2 \|G\|_2^2 \|H\|_2^2 + \|\xi\|^2 - 2\alpha \|H\|_2 \langle G, \xi \rangle$$
$$\leq n\alpha^2 \|H\|_2^2 \left(1 + 2\sqrt{\frac{\log(32/\delta)}{n}}\right)^2 + \|\xi\|^2 + 2\sqrt{n}\alpha \|H\|_2 \|\xi\|_2 \sqrt{\frac{2\log(16/\delta)}{n}}$$
$$\leq n\alpha^2 \|H\|_2^2 \left(1 + 10\sqrt{\frac{\log(32/\delta)}{n}}\right) + \left(1 + \sqrt{\frac{2\log(16/\delta)}{n}}\right) \|\xi\|_2^2$$

and it suffices to let

$$\alpha^2 \|H\|_2^4 \geq n\alpha^2 \|H\|_2^2 \left(1 + 10\sqrt{\frac{\log(32/\delta)}{n}}\right) + \left(1 + \sqrt{\frac{2\log(16/\delta)}{n}}\right) \|\xi\|_2^2.$$

Rearranging the above inequality, we can choose

$$\alpha = \left(\frac{1 + 10\sqrt{\frac{\log(32/\delta)}{n}}}{1 - \frac{n}{d_1 d_2}\left(1 + 10\sqrt{\frac{\log(32/\delta)}{n}}\right)\left(1 + 2\sqrt{\frac{\log(32/\delta)}{d_1 d_2}}\right)^2}\right)^{1/2} \frac{\sqrt{n\sigma^2}}{\|H\|_2^2}$$

and since $H$ as a matrix can have at most rank $d_1$, by Cauchy-Schwarz inequality on the singular values of $H$, we have $\|H\|_* \leq \sqrt{d_1}\|H\|_2$ and

$$\|x\|_* = \alpha \|H\|_* \leq \alpha \sqrt{d_1} \|H\|_2 \leq (1 + \epsilon)\sqrt{\frac{d_1(n\sigma^2)}{d_1 d_2}} = (1 + \epsilon)\sqrt{\frac{n\sigma^2}{d_2}}$$

for some $\epsilon \lesssim \sqrt{\frac{\log(32/\delta)}{n}} + \frac{n}{d_1 d_2}$. The desired conclusion follows by the observation that $\|X^*\|_* \leq \sqrt{r}\|X^*\|_F$ because $X^*$ has rank $r$. $\qquad\square$

**Theorem 5.** *Fix any $\delta \in (0,1)$. There exist constants $c_1, c_2, c_3 > 0$ such that if $d_1 d_2 > c_1 n$, $d_2 > c_2 d_1$, $n > c_3 r(d_1 + d_2)$, then with probability at least $1 - \delta$ that*

$$\frac{\|\hat{X} - X^*\|_F^2}{\|X^*\|_F^2} \lesssim \frac{r(d_1 + d_2)}{n} + \sqrt{\frac{r(d_1 + d_2)}{n}} \frac{\sigma}{\|X^*\|_F} + \left(\sqrt{\frac{d_1}{d_2}} + \frac{n}{d_1 d_2}\right) \frac{\sigma^2}{\|X^*\|_F^2}. \tag{18}$$

*Proof.* Note that $\langle A, X^* \rangle \sim \mathcal{N}(0, \|X^*\|_F^2)$ and so by the standard Gaussian tail bound $\Pr(|Z| \geq t) \leq 2e^{-t^2/2}$, Theorem 9 and a union bound, it holds with probability at least $1 - \delta/8$ that

$$|\langle A, X^* \rangle| \leq \sqrt{2 \log(32/\delta)} \|X^*\|_F$$
$$\|A\|_{op} \leq \sqrt{d_1} + \sqrt{d_2} + \sqrt{2 \log(32/\delta)}.$$

Then it holds that

$$\left\| A - \frac{\langle A, X^* \rangle}{\|X^*\|_F^2} X^* \right\|_{op} \leq \|A\|_{op} + \frac{|\langle A, X^* \rangle|}{\|X^*\|_F^2} \|X^*\|_{op}$$

$$\leq \sqrt{d_1} + \sqrt{d_2} + \sqrt{2 \log(32/\delta)} + \frac{\|X^*\|_{op}}{\|X^*\|_F} \sqrt{2 \log(32/\delta)}$$

$$\leq \sqrt{d_1} + \sqrt{d_2} + \sqrt{8 \log(32/\delta)}.$$

Therefore, we can choose $C_\delta$ in Theorem 1 by

$$C_\delta(X) := \left(\sqrt{d_1} + \sqrt{d_2} + \sqrt{8 \log(32/\delta)}\right) \|X\|_*$$

and applying Theorem 1 and Theorem 4, we have

$$(1 - \epsilon) L(\hat{X}) \leq \frac{C_\delta(X)^2}{n}$$

$$\leq \frac{\left(\sqrt{d_1} + \sqrt{d_2} + \sqrt{8 \log(32/\delta)}\right)^2}{n} \left(\sqrt{r} \|X^*\|_F + (1 + \epsilon) \sqrt{\frac{n \sigma^2}{d_1 \vee d_2}}\right)^2$$

$$= \left(\sqrt{\frac{d_1}{d_1 \vee d_2}} + \sqrt{\frac{d_2}{d_1 \vee d_2}} + \sqrt{\frac{8 \log(32/\delta)}{d_1 \vee d_2}}\right)^2 \left(\sqrt{\frac{r(d_1 \vee d_2)}{n}} + (1 + \epsilon) \frac{\sigma}{\|X^*\|_F}\right)^2 \|X^*\|_F^2$$

where $\epsilon$ is the maximum of the two $\epsilon$ in Theorem 1 and Theorem 4. Finally, recall that

$$L(\hat{X}) = \sigma^2 + \|\hat{X} - X^*\|_F^2.$$

Assuming that $d_1 \leq d_2$, then the above implies that

$$\frac{\|\hat{X} - X^*\|_F^2}{\|X^*\|_F^2}$$

$$\leq (1 - \epsilon)^{-1} (1 + \epsilon)^2 \left(1 + \sqrt{\frac{d_1}{d_2}} + \sqrt{\frac{8 \log(32/\delta)}{d_2}}\right)^2 \left(\sqrt{\frac{r(d_1 + d_2)}{n}} + \frac{\sigma}{\|X^*\|_F}\right)^2 - \frac{\sigma^2}{\|X^*\|_F^2}$$

$$\lesssim \frac{r(d_1 + d_2)}{n} + \sqrt{\frac{r(d_1 + d_2)}{n}} \frac{\sigma}{\|X^*\|_F} + \left(\sqrt{\frac{d_1}{d_2}} + \frac{n}{d_1 d_2}\right) \frac{\sigma^2}{\|X^*\|_F^2}$$

and we are done. $\qquad\square$

## E   Counterexample to Gaussian Universality

By assumption (G), we can write $x_{i|d-k} = h(x_{i|k}) \cdot \Sigma_{|d-k}^{1/2} z_i$ where $z_i \sim \mathcal{N}(0, I_{d-k})$. We will denote the matrix $Z = [z_1, ..., z_n]^T \in \mathbb{R}^{n \times (d-k)}$. Following the notation in section 7, we will also write $X = [X_{|k}, X_{|d-k}]$ where $X_{|k} \in \mathbb{R}^{n \times k}$ and $X_{|d-k} \in \mathbb{R}^{n \times (d-k)}$. The proofs in this section closely follows the proof of Theorem 6.

**Theorem 15.** *Consider dataset $(X, Y)$ drawn i.i.d. from the data distribution $\mathcal{D}$ according to (G) and (H), and fix any $f : \mathbb{R} \times \mathcal{Y} \to \mathbb{R}_{\geq 0}$ such that $\sqrt{f}$ is 1-Lipschitz for any $y \in \mathcal{Y}$. Fix any $\delta > 0$ and suppose there exists $\epsilon_\delta < 1$ and $C_\delta : \mathbb{R}^{d-k} \to [0, \infty]$ such that*

(i) *with probability at least $1 - \delta/2$ over $(X, Y)$ and $G \sim \mathcal{N}(0, I_n)$, it holds uniformly over all $w_{|k} \in \mathbb{R}^k$ and $\|w_{|d-k}\|_{\Sigma_{|d-k}} \in \mathbb{R}_{\geq 0}$ that*

$$\frac{1}{n} \sum_{i=1}^{n} \frac{f(\langle w_{|k}, x_{i|k} \rangle + h(x_{i|k}) \|w_{|d-k}\|_{\Sigma_{|d-k}} G_i, y_i)}{h(x_{i|k})^2} \geq (1 - \epsilon_\delta) \mathbb{E}_{\mathcal{D}} \left[ \frac{f(\langle w, x \rangle, y)}{h(x_{|k})^2} \right]$$

(ii) *with probability at least $1 - \delta/2$ over $z_{|d-k} \sim \mathcal{N}(0, \Sigma_{|d-k})$, it holds uniformly over all $w_{|d-k} \in \mathbb{R}^{d-k}$ that*

$$\langle w_{|d-k}, z_{|d-k} \rangle \leq C_\delta(w_{|d-k}) \tag{75}$$

*then with probability at least $1 - \delta$, it holds uniformly over all $w \in \mathbb{R}^d$ that*

$$(1 - \epsilon_\delta) \mathbb{E} \left[ \frac{f(\langle w, x \rangle, y)}{h(x_{|k})^2} \right] \leq \left( \frac{1}{n} \sum_{i=1}^{n} \frac{f(\langle w, x_i \rangle, y_i)}{h(x_{i|k})^2} + \frac{C_\delta(w_{|d-k})}{\sqrt{n}} \right)^2. \tag{76}$$

*Proof.* Note that

$$\langle w_{|d-k}, x_{i|d-k} \rangle = h(x_{i|k}) \cdot \langle w_{|d-k}, \Sigma_{|d-k}^{1/2} z_i \rangle$$

and so for any $f : \mathbb{R} \times \mathcal{Y} \times \mathbb{R}^k \to \mathbb{R}$, we can write

$$\Phi := \sup_{w \in \mathbb{R}^d} F(w) - \frac{1}{n} \sum_{i=1}^{n} f(\langle w, x_i \rangle, y_i, x_{i|k})$$

$$= \sup_{\substack{w \in \mathbb{R}^d, u \in \mathbb{R}^n \\ u = Z \Sigma_{|d-k}^{1/2} w_{|d-k}}} F(w) - \frac{1}{n} \sum_{i=1}^{n} f(\langle w_{|k}, x_{i|k} \rangle + h(x_{i|k}) u_i, y_i, x_{i|k})$$

$$= \sup_{w \in \mathbb{R}^d, u \in \mathbb{R}^n} \inf_{\lambda \in \mathbb{R}^n} \langle \lambda, Z \Sigma_{|d-k}^{1/2} w_{|d-k} - u \rangle + F(w) - \frac{1}{n} \sum_{i=1}^{n} f(\langle w_{|k}, x_{i|k} \rangle + h(x_{i|k}) u_i, y_i, x_{i|k}).$$

By the same truncation argument used in Lemma 7, it suffices to consider the auxiliary problem:

$$\Psi := \sup_{w \in \mathbb{R}^d, u \in \mathbb{R}^n} \inf_{\lambda \in \mathbb{R}^n} \|\lambda\|_2 \langle H, \Sigma_{|d-k}^{1/2} w_{|d-k} \rangle + \langle G \| \Sigma_{|d-k}^{1/2} w_{|d-k} \|_2 - u, \lambda \rangle$$

$$+ F(w) - \frac{1}{n} \sum_{i=1}^{n} f(\langle w_{|k}, x_{i|k} \rangle + h(x_{i|k}) u_i, y_i, x_{i|k})$$

$$= \sup_{w \in \mathbb{R}^d, u \in \mathbb{R}^n} \inf_{\lambda \geq 0} \lambda \left( \langle H, \Sigma_{|d-k}^{1/2} w_{|d-k} \rangle - \left\| G \| \Sigma_{|d-k}^{1/2} w_{|d-k} \|_2 - u \right\|_2 \right)$$

$$+ F(w) - \frac{1}{n} \sum_{i=1}^{n} f(\langle w_{|k}, x_{i|k} \rangle + h(x_{i|k}) u_i, y_i, x_{i|k})$$

Therefore, it holds that

$$\Psi = \sup_{\substack{w \in \mathbb{R}^d, u \in \mathbb{R}^n \\ \langle H, \Sigma_{|d-k}^{1/2} w_{|d-k} \rangle \geq \left\| G \| \Sigma_{|d-k}^{1/2} w_{|d-k} \|_2 - u \right\|_2}} F(w) - \frac{1}{n} \sum_{i=1}^{n} f(\langle w_{|k}, x_{i|k} \rangle + h(x_{i|k}) u_i, y_i, x_{i|k})$$

$$= \sup_{w \in \mathbb{R}^d} F(w) - \frac{1}{n} \inf_{\substack{u \in \mathbb{R}^n \\ \langle H, \Sigma_{|d-k}^{1/2} w_{|d-k} \rangle \geq \left\| G \| \Sigma_{|d-k}^{1/2} w_{|d-k} \|_2 - u \right\|_2}} \sum_{i=1}^{n} f(\langle w_{|k}, x_{i|k} \rangle + h(x_{i|k}) u_i, y_i, x_{i|k}).$$

Next, we analyze the infimum term:

$$\inf_{\substack{u\in\mathbb{R}^n \\ \langle H,\Sigma_{|d-k}^{1/2}w_{|d-k}\rangle\geq\left\|G\|\Sigma_{|d-k}^{1/2}w_{|d-k}\|_2-u\right\|_2}}\sum_{i=1}^{n}f(\langle w_{|k},x_{i|k}\rangle+h(x_{i|k})u_i,y_i,x_{i|k})$$

$$=\inf_{\substack{u\in\mathbb{R}^n \\ \|u\|_2\leq\langle H,\Sigma_{|d-k}^{1/2}w_{|d-k}\rangle}}\sum_{i=1}^{n}f\left(\langle w_{|k},x_{i|k}\rangle+h(x_{i|k})\left(u_i+\|\Sigma_{|d-k}^{1/2}w_{|d-k}\|_2G_i\right),y_i,x_{i|k}\right)$$

$$=\inf_{u\in\mathbb{R}^n}\sup_{\lambda\geq0}\lambda(\|u\|^2-\langle H,\Sigma_{|d-k}^{1/2}w_{|d-k}\rangle^2)$$

$$+\sum_{i=1}^{n}f\left(\langle w_{|k},x_{i|k}\rangle+h(x_{i|k})\left(u_i+\|\Sigma_{|d-k}^{1/2}w_{|d-k}\|_2G_i\right),y_i,x_{i|k}\right)$$

$$\geq\sup_{\lambda\geq0}\inf_{u\in\mathbb{R}^n}\lambda(\|u\|^2-\langle H,\Sigma_{|d-k}^{1/2}w_{|d-k}\rangle^2)$$

$$+\sum_{i=1}^{n}f\left(\langle w_{|k},x_{i|k}\rangle+h(x_{i|k})\left(u_i+\|\Sigma_{|d-k}^{1/2}w_{|d-k}\|_2G_i\right),y_i,x_{i|k}\right)$$

$$=\sup_{\lambda\geq0}-\lambda\langle H,\Sigma_{|d-k}^{1/2}w_{|d-k}\rangle^2$$

$$+\sum_{i=1}^{n}\inf_{u_i\in\mathbb{R}}f(\langle w_{|k},x_{i|k}\rangle+u_i+\|\Sigma_{|d-k}^{1/2}w_{|d-k}\|_2h(x_{i|k})G_i,y_i,x_{i|k})+\frac{\lambda}{h(x_{i|k})^2}u_i^2.$$

Now suppose that $f$ takes the form $f(\hat{y},y,x_{|k})=\frac{1}{h(x_{|k})^2}\tilde{f}(\hat{y},y)$ for some 1 square-root Lipschitz $\tilde{f}$ and by a union bound, it holds with probability at least $1-\delta$ that

$$\langle\Sigma_{|d-k}^{1/2}H,w_{|d-k}\rangle^2\leq C_\delta(w_{|d-k})^2$$

$$\frac{1}{n}\sum_{i=1}^{n}\frac{1}{h(x_{i|k})^2}\tilde{f}(\langle w_{|k},x_{i|k}\rangle+\|\Sigma_{|d-k}^{1/2}w_{|d-k}\|_2h(x_{i|k})G_i,y_i)\geq(1-\epsilon_\delta)\mathbb{E}\left[\frac{1}{h(x_{|k})^2}\tilde{f}(\langle w,x\rangle,y)\right],$$

then the above becomes

$$\sup_{\lambda\geq0}-\lambda\langle\Sigma_{|d-k}^{1/2}H,w_{|d-k}\rangle^2+\sum_{i=1}^{n}\frac{1}{h(x_{i|k})^2}\tilde{f}_\lambda(\langle w_{|k},x_{i|k}\rangle+\|\Sigma_{|d-k}^{1/2}w_{|d-k}\|_2h(x_{i|k})G_i,y_i)$$

$$\geq\sup_{\lambda\geq0}-\lambda\langle\Sigma_{|d-k}^{1/2}H,w_{|d-k}\rangle^2+\frac{\lambda}{\lambda+1}\sum_{i=1}^{n}\frac{1}{h(x_{i|k})^2}\tilde{f}(\langle w_{|k},x_{i|k}\rangle+\|\Sigma_{|d-k}^{1/2}w_{|d-k}\|_2h(x_{i|k})G_i,y_i)$$

$$\geq\sup_{\lambda\geq0}-\lambda C_\delta(w_{|d-k})^2+\frac{\lambda}{\lambda+1}(1-\epsilon)n\mathbb{E}\left[\frac{1}{h(x_{|k})^2}\tilde{f}(\langle w,x\rangle,y)\right]$$

$$\geq n\left(\sqrt{(1-\epsilon_\delta)\mathbb{E}\left[\frac{1}{h(x_{|k})^2}\tilde{f}(\langle w,x\rangle,y)\right]}-\frac{C_\delta(w_{|d-k})}{\sqrt{n}}\right)_+^2$$

where we apply Lemma 8 in the last step. Then if we take

$$F(w)=\left(\sqrt{(1-\epsilon_\delta)\mathbb{E}\left[\frac{1}{h(x_{|k})^2}\tilde{f}(\langle w,x\rangle,y)\right]}-\frac{C_\delta(w_{|d-k})}{\sqrt{n}}\right)_+^2$$

then we have $\Psi\leq0$. To summarize, we have shown

$$\left(\sqrt{(1-\epsilon_\delta)\mathbb{E}\left[\frac{1}{h(x_{|k})^2}\tilde{f}(\langle w,x\rangle,y)\right]}-\frac{C_\delta(w_{|d-k})}{\sqrt{n}}\right)_+^2-\frac{1}{n}\sum_{i=1}^{n}\frac{1}{h(x_{i|k})^2}\tilde{f}(\langle w,x_i\rangle,y_i)\leq0$$

which implies

$$\mathbb{E}\left[\frac{1}{h(x_{|k})^2}\tilde{f}(\langle w,x\rangle,y)\right]\leq(1-\epsilon_\delta)^{-1}\left(\frac{1}{n}\sum_{i=1}^{n}\frac{1}{h(x_{i|k})^2}\tilde{f}(\langle w,x_i\rangle,y_i)+\frac{C_\delta(w_{|d-k})}{\sqrt{n}}\right)^2.\quad\square$$

**Theorem 16.** *Under assumptions (G) and (H), fix any $w_{|k}^* \in \mathbb{R}^k$ and suppose for some $\rho \in (0,1)$, it holds with probability at least $1 - \delta/8$*

$$\frac{1}{n}\sum_{i=1}^n \left(\frac{y_i - \langle w_{|k}^*, x_{i|k}\rangle}{h(x_{i|k})}\right)^2 \le (1+\rho) \cdot \mathbb{E}\left[\left(\frac{y - \langle w_{|k}^*, x_{|k}\rangle}{h(x_{|k})}\right)^2\right]. \tag{77}$$

*Then with probability at least $1 - \delta$, for some $\epsilon \lesssim \rho + \log\left(\frac{1}{\delta}\right)\left(\frac{1}{\sqrt{n}} + \frac{1}{\sqrt{R(\Sigma_{|d-k})}} + \frac{n}{R(\Sigma_{|d-k})}\right)$, it holds that*

$$\min_{w\in\mathbb{R}^d:\forall i,\langle w,x_i\rangle=y_i} \|w\|_2^2 \le \|w_{|k}^*\|_2^2 + (1+\epsilon)\frac{n\mathbb{E}\left[\left(\frac{y-\langle w_{|k}^*, x_{|k}\rangle}{h(x_{|k})}\right)^2\right]}{\mathrm{Tr}(\Sigma_{|d-k})} \tag{78}$$

*Proof.* Fix any $w_{|k}^* \in \mathbb{R}^k$, we observe that

$$\min_{w\in\mathbb{R}^d:\forall i,\langle w,x_i\rangle=y_i} \|w\|_2^2 = \min_{w\in\mathbb{R}^d:\forall i,\langle w_{|k},x_{i|k}\rangle+\langle w_{|d-k},x_{i|d-k}\rangle=y_i} \|w_{|k}\|_2^2 + \|w_{|d-k}\|_2^2$$

$$\le \|w_{|k}^*\|_2^2 + \min_{\substack{w_{|d-k}\in\mathbb{R}^{d-k}:\\ \forall i,\langle w_{|d-k},x_{i|d-k}\rangle=y_i-\langle w_{|k}^*,x_{i|k}\rangle}} \|w_{|d-k}\|_2^2.$$

Therefore, it is enough analyze

$$\Phi := \min_{\substack{w_{|d-k}\in\mathbb{R}^{d-k}:\\ \forall i,\langle w_{|d-k},x_{i|d-k}\rangle=y_i-\langle w_{|k}^*,x_{i|k}\rangle}} \|w_{|d-k}\|_2 = \min_{\substack{w_{|d-k}\in\mathbb{R}^{d-k}:\\ \forall i,\langle w_{|d-k},\Sigma_{|d-k}^{1/2}z_i\rangle=\frac{y_i-\langle w_{|k}^*,x_{i|k}\rangle}{h(x_{i|k})}}} \|w_{|d-k}\|_2.$$

By introducing the Lagrangian, we have

$$\Phi = \min_{w_{|d-k}\in\mathbb{R}^{d-k}} \max_{\lambda\in\mathbb{R}^n} \sum_{i=1}^n \lambda_i\left(\langle \Sigma_{|d-k}^{1/2}w_{|d-k}, z_i\rangle - \frac{y_i - \langle w_{|k}^*, x_{i|k}\rangle}{h(x_{i|k})}\right) + \|w_{|d-k}\|_2$$

$$= \min_{w_{|d-k}\in\mathbb{R}^{d-k}} \max_{\lambda\in\mathbb{R}^n} \langle \lambda, Z\Sigma_{|d-k}^{1/2}w_{|d-k}\rangle - \sum_{i=1}^n \lambda_i\left(\frac{y_i - \langle w_{|k}^*, x_{i|k}\rangle}{h(x_{i|k})}\right) + \|w_{|d-k}\|_2.$$

Similarly, the above is only random in $Z$ after conditioning on $X_{|k}w_{|k}^*$ and $\xi$ and the distribution of $Z$ remains unchanged after conditioning because of the independence. By the same truncation argument as before and CGMT, it suffices to consider the auxiliary problem:

$$\min_{w_{|d-k}\in\mathbb{R}^{d-k}} \max_{\lambda\in\mathbb{R}^n} \|\lambda\|_2\langle H, \Sigma_{|d-k}^{1/2}w_{|d-k}\rangle + \sum_{i=1}^n \lambda_i\left(\|\Sigma_{|d-k}^{1/2}w_{|d-k}\|_2 G_i - \frac{y_i - \langle w_{|k}^*, x_{i|k}\rangle}{h(x_{i|k})}\right)$$
$$+ \|w_{|d-k}\|_2$$

$$= \min_{w_{|d-k}\in\mathbb{R}^{d-k}} \max_{\lambda\in\mathbb{R}^n} \|\lambda\|_2\left(\langle H, \Sigma_{|d-k}^{1/2}w_{|d-k}\rangle + \sqrt{\sum_{i=1}^n\left(\|\Sigma_{|d-k}^{1/2}w_{|d-k}\|_2 G_i - \frac{y_i - \langle w_{|k}^*, x_{i|k}\rangle}{h(x_{i|k})}\right)^2}\right)$$
$$+ \|w_{|d-k}\|_2$$

and so we can define

$$\Psi := \min_{\substack{w_{|d-k}\in\mathbb{R}^{d-k}:\\ \sqrt{\sum_{i=1}^n\left(\|\Sigma_{|d-k}^{1/2}w_{|d-k}\|_2 G_i - \frac{y_i-\langle w_{|k}^*,x_{i|k}\rangle}{h(x_{i|k})}\right)^2}\le\langle-\Sigma_{|d-k}^{1/2}H, w_{|d-k}\rangle}} \|w_{|d-k}\|_2.$$

To upper bound $\Psi$, we consider $w_{|d-k}$ of the form $-\alpha\frac{\Sigma_{|d-k}^{1/2}H}{\|\Sigma_{|d-k}^{1/2}H\|_2}$, then we just need

$$\sum_{i=1}^n\left(\alpha\frac{\|\Sigma_{|d-k}H\|_2}{\|\Sigma_{|d-k}^{1/2}H\|_2}G_i - \frac{y_i - \langle w_{|k}^*, x_{i|k}\rangle}{h(x_{i|k})}\right)^2 \le \alpha^2\|\Sigma_{|d-k}^{1/2}H\|_2^2.$$

By a union bound, the following occur together with probability at least $1 - \delta/2$ for some absolute constant $C > 0$:

1. Using the first part of Lemma 4, we have

$$\|\Sigma_{|d-k}^{1/2} H\|_2^2 \geq \operatorname{Tr}(\Sigma_{|d-k}) \left(1 - C \frac{\log(32/\delta)}{\sqrt{R(\Sigma_{|d-k})}}\right)$$

2. Using the last part of Lemma 4, requiring $R(\Sigma_{|d-k}) \gtrsim \log(32/\delta)^2$

$$\frac{\|\Sigma_{|d-k} H\|_2^2}{\|\Sigma_{|d-k}^{1/2} H\|_2^2} \leq C \log(32/\delta) \frac{\operatorname{Tr}(\Sigma_{|d-k}^2)}{\operatorname{Tr}(\Sigma_{|d-k})}$$

3. Using subexponential Bernstein's inequality (Theorem 2.8.1 of Vershynin (2018)), requiring $n = \Omega(\log(1/\delta))$,

$$\frac{1}{n} \sum_{i=1}^{n} G_i^2 \leq 2$$

4. Using standard Gaussian tail bound $\Pr(|Z| \geq t) \leq 2e^{-t^2/2}$, we have

$$\left| \frac{1}{n} \sum_{i=1}^{n} \frac{G_i(y_i - \langle w_{|k}^*, x_{i|k} \rangle)}{h(x_{i|k})} \right| \leq \sqrt{\frac{1}{n} \sum_{i=1}^{n} \left( \frac{y_i - \langle w_{|k}^*, x_{i|k} \rangle}{h(x_{i|k})} \right)^2} \sqrt{\frac{2\log(32/\delta)}{n}}$$

5. By assumption, it holds that

$$\frac{1}{n} \sum_{i=1}^{n} \left( \frac{y_i - \langle w_{|k}^*, x_{i|k} \rangle}{h(x_{i|k})} \right)^2 \leq (1 + \rho) \cdot \mathbb{E}\left[ \left( \frac{y - \langle w_{|k}^*, x_{|k} \rangle}{h(x_{|k})} \right)^2 \right].$$

Then we use the above and the AM-GM inequality to show that

$$\frac{1}{n} \sum_{i=1}^{n} \left( \alpha \frac{\|\Sigma_{|d-k} H\|_2}{\|\Sigma_{|d-k}^{1/2} H\|_2} G_i - \frac{y_i - \langle w_{|k}^*, x_{i|k} \rangle}{h(x_{i|k})} \right)^2$$

$$\leq 2\alpha^2 \frac{\|\Sigma_{|d-k} H\|_2^2}{\|\Sigma_{|d-k}^{1/2} H\|_2^2} + (1 + \rho) \cdot \mathbb{E}\left[ \left( \frac{y - \langle w_{|k}^*, x_{|k} \rangle}{h(x_{|k})} \right)^2 \right]$$

$$+ 2 \frac{\alpha \|\Sigma_{|d-k} H\|_2}{\|\Sigma_{|d-k}^{1/2} H\|_2} \sqrt{(1 + \rho) \cdot \mathbb{E}\left[ \left( \frac{y - \langle w_{|k}^*, x_{|k} \rangle}{h(x_{|k})} \right)^2 \right]} \sqrt{\frac{2\log(32/\delta)}{n}}$$

$$\leq C \log(32/\delta) \left( 2 + \sqrt{\frac{2\log(32/\delta)}{n}} \right) \alpha^2 \frac{\operatorname{Tr}(\Sigma_{|d-k}^2)}{\operatorname{Tr}(\Sigma_{|d-k})}$$

$$+ \left( 1 + \sqrt{\frac{2\log(32/\delta)}{n}} \right) (1 + \rho) \cdot \mathbb{E}\left[ \left( \frac{y - \langle w_{|k}^*, x_{|k} \rangle}{h(x_{|k})} \right)^2 \right].$$

After some rearrangements, it is easy to see that we can choose

$$\alpha^2 = \frac{\left( 1 + \sqrt{\frac{2\log(32/\delta)}{n}} \right)(1 + \rho)}{1 - C \frac{\log(32/\delta)}{\sqrt{R(\Sigma_{|d-k})}} - C\log(32/\delta) \left( 2 + \sqrt{\frac{2\log(32/\delta)}{n}} \right) \frac{n}{R(\Sigma_{|d-k})}} \cdot \frac{n\mathbb{E}\left[ \left( \frac{y - \langle w_{|k}^*, x_{|k} \rangle}{h(x_{|k})} \right)^2 \right]}{\operatorname{Tr}(\Sigma_{|d-k})}.$$

and the proof is complete. $\qquad\square$

