# OpenReview forum: "Uniform Convergence with Square-Root Lipschitz Loss"
_NeurIPS.cc/2023/Conference — NeurIPS 2023 poster_

### Official Review · Reviewer_Y88d · 2023-07-05

**Soundness:** 4 excellent
**Presentation:** 4 excellent
**Contribution:** 3 good
**Rating:** 7
**Confidence:** 1

**Summary:**

This paper develops uniform convergence result for empirical risk minimization with a loss function whose square root is Lipschitz. It is assumed that the covariate vector $x$ is drawn from a $d$-dimensional multivariate normal and the response $y$ is generated by a structural equation that only depends on $x$ through its $k$-dimensional projection. The result bounds the root of population risk in terms of the root of the empirical risk and another term concerning the Lipschitz constant, a complexity measure and the sample size. The bound holds uniformly (in high probability) over all possible affine coefficients in the empirical risk minimization. This key result is then applied to study "benign overfitting" in several problems, including phase retrieval, ReLU regression and matrix sensing. In addition, it is also shown that Gaussian universality does not hold for the setting considered.

**Strengths:**

1. The paper builds upon and generalizes related results in the literature.
2. It establishes a key generic result and then illustrates its strength through application to several topical problems.
3. The presentation is rigorous and clear.

**Weaknesses:**

As this paper falls out of my area of expertise, I am only listing below a few points that might help me (or someone from outside the field) to better understand the paper.

1. The notion of "consistency" or a "consistent loss" is mentioned in the Introduction and throughout the applications. It is worth explaining what that means and how it relates the usual notion of "consistency" in statistics.
2. Assumption (C) perhaps deserves more explanation: why is it necessary and when is it expected to hold?


**Questions:**

1. Page 4, line 135: I do not see why the distribution of $\langle w, x \rangle$ only depends on $w^{T} \Sigma W$. By $x \sim \mathcal{N}(\mu, \Sigma)$ as in Assumption (A), wouldn't it depend on $w^{T} \mu$ and $w^T \Sigma w$?
2. Page 2, line 76: missing "be" before "applied".

**Limitations:**

I do not foresee any potential negative societal impact of this work.

---

> ### Author Rebuttal · Authors · 2023-08-09
>
> Thanks for your feedback!
>
> > The notion of "consistency" or a "consistent loss" is mentioned in the Introduction and throughout the applications. It is worth explaining what that means and how it relates the usual notion of "consistency" in statistics.
>
> We say that an estimator (\hat{w},\hat{b}) valued in a set K is consistent if the test error of the estimator converges to the optimal test error in the class, i.e.
> L(\hat{w}, \hat{b}) -> \min_{w,b \in K} L(w,b).
>
> Here when we take the limit, we are considering an asymptotic regime where the number of samples n goes to infinity but also the dimension and other parameters of the problem may be assumed to scale in a certain way as well. (In classical parametric statistics, consistency was largely studied where the dimension was fixed and number of samples goes to infinity. In high-dimensional statistics like this work, the cited work of Bartlett et al, Thrampoulidis et al, etc. the dimension also goes to infinity with the number of samples and this is crucial to observe modern phenomena like benign overfitting.) For e.g. phase retrieval we show consistency under the same asymptotic conditions on the covariance matrix \Sigma that Bartlett et al. studied for linear regression.
>
> A key point with the above definition is that consistency is not just a property of the estimator, but also a property of the loss function f chosen. When we study benign overfitting in linear regression, phase retrieval, etc the estimator we are interested in will interpolate the data, and it is not clear a priori which losses this estimator will be consistent under (if any). The previous works on benign overfitting showed that in linear regression, minimum-norm interpolants will be consistent under the squared loss (but not e.g. the L1 loss if the model is misspecified). A major new contribution of this work is to prove consistency results for phase retrieval, ReLU regression etc and this requires in particular us to identify the consistent loss.
>
> > Assumption (C) perhaps deserves more explanation: why is it necessary and when is it expected to hold?
>
> Assumption ( C ) is a common assumption in the statistical learning theory literature which goes by a few different names, such as hypercontractivity or ‘norm equivalence’. To understand this assumption, it helps to think of a simple example, so consider for a moment linear regression with the usual squared loss and Gaussian noise & covariates. Then the assumption is saying that $E[(Y - <w, X>)^8] <= \tau E[(Y - <w, X>)^2]^4$ for any predictors w. Since the law of Y - <w, X> is just a Gaussian distribution with a certain variance, this is true simply because for a Gaussian random variable Z, it satisfies E[Z^8] <= 105 E[Z^2]^4.
>
> More generally, hypercontractivity will certainly be true for the squared loss if the class of functions is subgaussian in the sense of [Lecue-Mendelson ‘13] (https://arxiv.org/abs/1305.4825), and in fact hypercontractivity is a weaker assumption since it doesn’t require the existence of arbitrarily large moments of the distribution.
>
> We want to emphasize that we are not proposing hypercontractivity as a new assumption, we are simply using it as an existing and well-known assumption which makes dealing with the ‘low-dimensional concentration’ part of our analysis clean and straightforward. Because the original work of [Vapnik ‘82] studied the same assumption, we can directly cite his results in our analysis. However, as discussed in Appendix B.3 and [Zhou et al ‘22], it is also possible to apply other results from statistical learning theory to handle the low-dimensional concentration part of the argument and this would yield different versions of the main theorem.
>
> _Some_ type of concentration/anticoncentration assumption must always be made for any nonasymptotic guarantees on the test error to be possible, simply to avoid degenerate situations. For example, consider two cases: (1) Y = 0 always, or (2)  the true Y equals 0 with probability 1 - \xi and equals 1/\xi otherwise. For \xi -> 0 with a finite number of samples, we will not observe a sample where Y is nonzero, so we cannot distinguish situations (1) and (2). However, in situation (1) 0 is a perfect predictor of Y, whereas in situation (2) it suffers a very large squared loss. Standard assumptions like boundedness or hypercontractivity fix the problem because they rule out situation (2).
>
>
>
> > Page 4, line 135: I do not see why the distribution of ⟨�,�⟩ only depends on ��Σ�. By
> �∼�(�,Σ) as in Assumption (A), wouldn't it depend on ��� and ��Σ�?
>
> You are right, this is a typo and it should also depend on $w^T \mu$ and $w^T \Sigma w$. The point is that it only depends on these O(k) many quantities instead of O(d) many.

---

> > ### Comment · Reviewer_Y88d · 2023-08-14
> >
> > I want to thank the authors for addressing all my concerns!

---

### Official Review · Reviewer_K77B · 2023-07-05

**Soundness:** 4 excellent
**Presentation:** 3 good
**Contribution:** 3 good
**Rating:** 6
**Confidence:** 3

**Summary:**

The paper introduces sharp uniform convergence guarantees for generalized linear models in Gaussian space  for square-root Lipschitz losses. These results extend the scope of previous findings and open up possibilities for applying the derived loss bounds in new contexts.

**Strengths:**

1.The paper is well-written, providing clear explanations of its contributions in relation to previous work.
2.The paper achieves fast convergence rates for a broader class of losses, surpassing previous research in this area.
3.By simplifying the assumptions compared to previous work, the paper is able to derive new bounds applicable to scenarios where solving the Moreau envelope of the loss function is challenging or not possible in closed-form.

**Weaknesses:**

While the authors discuss potential extensions of the Gaussian feature assumption,  the paper still focuses on the setting of Gaussian data, similar to previous work. Additionally, the proof technique employed in the paper is not entirely novel, as it follows prior works that utilize the Gaussian Minimax Theorem for establishing uniform convergence.

**Questions:**

See weaknesses.

**Limitations:**

yes

---

> ### Author Rebuttal · Authors · 2023-08-09
>
> Thanks for your feedback! We would like to discuss two points from your comments: (1) the Gaussianity assumption and (2) the novelty of the proof techniques.
>
> *Regarding Gaussianity.* The Gaussianity assumption is indeed a significant restriction, and deviates from much of the classical distribution-free statistical learning literature.  But assuming Gaussianity allows obtaining much tighter guarantees, with tight numerical constants (see [Koehler et al ‘21, Zhou et al ‘21]).  Gaussianity is also widely assumed and used in analysis of many statistical learning and inference problems, e.g. sparse recovery [e.g. Stojnic ‘13, Chandrasekaran et al ‘12 https://arxiv.org/abs/1012.0621, …], phase retrieval [e.g. Mondelli and Montanari ‘18, Barbier et al ‘19], logistic regression [e.g. Candes-Sur ‘20 https://arxiv.org/abs/1804.09753], many other works such as those using Approximate Message Passing, and even analysis of deep learning [e.g. Soltanolkotabi ‘17 https://arxiv.org/abs/1705.04591] (although most of these were also studied, usually with weaker guarantees, without assuming Gaussianity).  Furthermore, much of the statistical physics based analysis relies on “Gaussian universality”, which essentially also relies on the data behaving as if it was Gaussian [e.g. see Hu-Lu ‘23, Bayati et al ‘12 https://arxiv.org/pdf/1207.7321.pdf for some discussion and related rigorous results].  And so, although we agree this is a significant restriction, and it would be very interesting to relax this assumption, we still believe that results relying on Gaussianity are interesting and useful, both on their own right, and as a step toward more general analysis.
>
> Specifically in the context of this work, by working with Gaussian data we were able to discover several interesting phenomena (e.g. benign overfitting in ReLU regression) which were not at all obvious beforehand (for example, that the consistent loss for interpolation in ReLU regression is (13), that this loss is sqrt-lipschitz, and that it satisfies an “optimistic rate” bound). With a view towards understanding all of these new phenomena beyond the Gausian setting, we have included Section 7, which reveals a situation where a naive generalization of our optimistic rates bound to non-gaussian data is false, but a correct & more sophisticated generalization (equation (28)) works.
>
> *Regarding the novelty of the analysis.* Besides the proof of the main generalization bound (which has some new elements, see reply to reviewer 8w1y), there is a very substantial amount of technical content contained within sections 5-7 which goes well beyond the previous literature. As a reminder, a very impressive theoretical understanding of benign overfitting in linear models (e.g. kernel machines) has emerged over the past few years. But studying (even mildly) nonlinear models has been a significant challenge since they do not seem as amenable to random matrix theory methods. One of the most important (and surprising!) realizations we had in this work was how to analyze benign overfitting in nonlinear models like phase retrieval & ReLU regression. At a technical level, this comes from the discovery of 1. the construction described in equation (8) which gives a low-norm interpolator for these nonlinear models, and 2. the closely related discovery of the correct ‘consistent losses’ for these problems. This was not at all obvious a priori! (E.g. we are not aware of anybody discovering the consistent loss (13) for interpolating ReLU regression before this work, or realizing that the Bartlett et al conditions for linear regression should also be sufficient for ReLU regression.) The extensions of the theory to matrix sensing, simple neural networks, and the non-gaussian setting of section 7 are also serious new contributions to the literature which cannot be obtained from previous work. In summary, the fact that sqrt-lipshitz losses satisfy a very sharp optimistic rates bound was one key finding, but the fact that combining this bound with the _right_ choice of sqrt-lipschitz losses lets us understand so many new phenomena is the deeper conceptual message of this work.

---

> > ### Comment · Reviewer_K77B · 2023-08-17
> > **Answer to the rebuttal**
> >
> > Thanks to the authors for addressing my concerns!

---

### Official Review · Reviewer_JChX · 2023-07-23

**Soundness:** 3 good
**Presentation:** 2 fair
**Contribution:** 3 good
**Rating:** 4
**Confidence:** 3

**Summary:**

This paper investigates optimistic rates under square-root Lipschitz losses and Gaussian data. Applications to phase retrieval, ReLu regression, matrix sensing, and single-index NNs are given.


**Strengths:**

The paper extends the analysis of optimistic rates to square-root Lipschitz losses, which goes beyond the classical smoothness assumption, which is interesting.

**Weaknesses:**

The authors claim that the obtained results provide a better understanding of benign overfitting, but the paper needs more explanations and comments on the main results.

As an example, the bound of eq. (10) features ten distinct terms. It would be nice to discuss which terms refer to what and how they compare with existing terms.

Another aspect I need clarification on is the notion of uniform convergence. In traditional statical learning, uniform convergence controls the rate of decay of |R(f) - \hat{R}_n(f)| for all f \in F and every distribution in a given class. On the other hand, the rate in Th.1 feature (1-\varepsilon) R(f) on the LHS. I am unfamiliar with benign overfitting literature (which could be standard). However, I invite the authors to add a discussion on this to broaden the paper's audience.

As for the assumption, I think the Gaussian data is quite restrictive. In modern applications, it is not uncommon to have abundant data but poor quality. In these cases, the data distribution feature tails fatter than any sub-Gaussian (and even sub-Exponential).

Assumption (C) is related to the L_4 norm of the loss. It looks formidable to verify in practice. Even for sub-Gaussian losses, the L_4 would be proportional to \sqrt{4}*\sqrt{VAR(loss)}, while the condition requires proportionality to the E[loss].

Please provide a discussion on these aspects. I may raise my score accordingly.

**Questions:**

Please provide more discussion on the highlighted weaknesses.

**Limitations:**

Assuming the data are Gaussian is limiting.

Please provide examples when Assumption (C) is satisfied.

---

> ### Author Rebuttal · Authors · 2023-08-09
>
> Thanks for your feedback and questions!
>
> > As an example, the bound of eq. (10) features ten distinct terms... It would be nice to discuss which terms refer to what...
>
> The term \rho corresponds to the deviation between the train error and test error of the reference predictor w^# . The terms appearing on the rhs of the inequality for \epsilon are small assuming Bartlett et al’s “benign overfitting conditions” on the covariance matrix \Sigma. Basically, Bartlett et al’s conditions require that \Sigma can be split into a low rank part and an orthogonal component with small trace and large effective rank (R(\Sigma^{\perp})). The parameter k lets us control the split between the low-rank part and the rest of \Sigma.
>
> Though the resulting rhs of equation (10) may appear mysterious at first sight, once we combine it into the generalization inequality to get equation (11) there is a lot of cancellation and we can see that the rhs of (10) is exactly the right size to guarantee consistency under benign overfitting conditions. We also see that the dependencies involved with \epsilon were necessary, because if benign overfitting conditions aren’t satisfied we shouldn’t be able to arrive at equation (11) (the conditions of Bartlett et al are close to necessary for benign overfitting, besides being sufficient).
>
> > Another aspect I need clarification on is the notion of uniform convergence... the rate in Th.1 feature (1-\varepsilon) R(f) on the LHS ...
>
> The (1 - \varepsilon) factor in the bound is common in statistical learning theory — it is an elegant way to state uniform convergence bounds for classes of functions where the risk R(f) can vary over different scales. If we wanted to, we could rearrange the bound to be of the form R(f) - \hat{R}_n(f) <= \varepsilon R(f) + … and if we assume an a priori bound on R(f) (e.g. assume the class of functions is bounded) this bound will be exactly of the form you suggest. Stating the inequality with the \varepsilon R(f) term is better, because as we obtain better upper bounds on R(f) the guarantee from the rhs improves. (This is related to “localization” in statistical learning.)
>
> As far as the historical origin, if you look at Theorem 13 in the appendix (which is from Vapnik 1982) you can see that it has a (1 - \varepsilon) factor in the same way on the rhs. Or for another example, the main result in “Extending the scope of the small-ball method” [Mendelson ‘20] also has such a factor and that paper has a lot of interesting discussion about these types of things.
>
> > As for the assumption, I think the Gaussian data is quite restrictive. In modern applications, it is not uncommon to have abundant data but poor quality. In these cases, the data distribution feature tails fatter than any sub-Gaussian (and even sub-Exponential).
>
> We agree that assuming the data is Gaussian is restrictive and it is a great direction for future work. Assuming Gaussianity is often a very helpful first step for proving more general results in this area — first we figure out what is true in the Gaussian case, and then we can try to extend it to more general distributions. For this same reason, we included the discussion in Section 7 to illustrate a situation where this type of universality fails, which we hope will help guide future research. (See the response to reviewer 949a for more discussion.)
>
> In [Zhou et al ‘22], there are related experimental results supporting the belief that sharp generalization theory from the Gaussian case should have more broadly applicable analogues — these experiments include very heavy-tailed distributions like the ones you mention.
>
> > Assumption (C) is related to the L_4 norm of the loss...  Please provide examples when Assumption (C) is satisfied.
>
> Assumption ( C ) is a common assumption in the statistical learning theory literature which goes by a few different names, such as hypercontractivity or ‘norm equivalence’. To understand this assumption, it helps to think of a simple example, so consider for a moment linear regression with the usual squared loss and Gaussian noise & covariates. Then the assumption is saying that $E[(Y - <w, X>)^8] <= \tau E[(Y - <w, X>)^2]^4$ for any predictors w. Since the law of Y - <w, X> is a Gaussian distribution with a certain variance, this is true simply because for a Gaussian random variable Z, it satisfies E[Z^8] <= 105 E[Z^2]^4.
>
> More generally, hypercontractivity will` be true for the squared loss if the class of functions is subgaussian in the sense of [Lecue-Mendelson ‘13] (https://arxiv.org/abs/1305.4825). Hypercontractivity is a weaker assumption since it doesn’t require the existence of arbitrarily large moments of the distribution.
>
> We want to emphasize that we are not proposing hypercontractivity as a new assumption — it is an existing and well-known assumption which makes dealing with the ‘low-dimensional concentration’ part of our analysis clean and straightforward. Because the original work of [Vapnik ‘82] studied the same assumption, we can directly cite his results. However, as discussed in Appendix B.3 and [Zhou et al ‘22], it is also possible to apply other results from statistical learning theory to handle the low-dimensional concentration part of the argument and this would yield different versions of the main theorem.
>
> _Some_ type of concentration/anticoncentration assumption must always be made for any nonasymptotic guarantees on the test error to be possible. For example, consider two cases: (1) Y = 0 always, or (2)  the true Y equals 0 with probability 1 - \xi and equals 1/\xi otherwise. For \xi -> 0 with a finite number of samples, we will not observe a sample where Y is nonzero, so we cannot distinguish situations (1) and (2). However, in situation (1) 0 is a perfect predictor of Y, whereas in situation (2) it suffers a very large squared loss. Standard assumptions like boundedness or hypercontractivity fix the problem because they rule out situation (2).

---

> > ### Comment · Reviewer_JChX · 2023-08-17
> >
> > I thanks the authors for addressing some of my concerns, especially regarding the hyper-contractivity.

---

### Official Review · Reviewer_8w1y · 2023-07-25

**Soundness:** 3 good
**Presentation:** 2 fair
**Contribution:** 2 fair
**Rating:** 7
**Confidence:** 3

**Summary:**

In the paper, the authors extended a type of sharp uniform convergence guarantee for the square loss to any square-root Lipschitz loss. The proof is considered to be simplified compared to ...

In the paper, the authors extend the theory of optimistic rates, a type of sharp uniform convergence guarantee, to square-root-Lipschitz losses, enabling new applications like phase retrieval. The authors have shown the uniform convergence for the multi-index model with Gaussian feature when the loss function is non-negative, square-root Lipshichtz and satisfies hypercontractivity. The convergence is controlled by the Radamacher complexity of the hypothesis class and the square-root Lipshichitz constant. The authors then show how this result can be applied to applications including phase retrieval, ReLU regression, matrix sensing, and single-index neural networks.

Compared to other loss function classes being considered in the literature for uniform convergence, square-root Lipshichtz loss is more general that it includes certain nonsmooth non-convex functions, and provides a better intuition of where the square root comparison between loss functions shows up.

The author also provides a counter-example to argue that Gaussian universality cannot always be taken for granted, and why the result can be over-optimistic for non-Gaussian data.

**Strengths:**

1. This paper goes beyond Lipschitz and smooth loss functions and extends the analysis to square-root-Lipschitz losses. This extension is valuable as it includes square loss and can address nonsmoothness. Also, it is easier to verify compared to the Moreau envelope condition used in [Zhou et al. 2022]. As the authors have shown in section 5, their results can be applied to show the benign overfitting for a wide range of real-world scenarios, yielding consistent results compared to past literature.

2. The concept of square-root Lipschitz losses is clean to state, and it indeed provides good intuition for the square-root relationship between losses shown up in previous optimistic rates analysis ([Zhou et al. 2020]). As a result, this new loss function class does seem to capture some intrinsic principle of the problem of optimistic rate and does contribute intuitions for the community. This expansion of applicable loss functions enhances the paper’s relevance and potential impact.

3. Application in Neural Networks: The paper’s extension of norm-based bounds and optimistic rates to weight-tied neural networks contributes to the field of neural network research. By providing a generalization bound that can be combined with the algorithmic outputs of Bietti et al., the authors offer practical insights for optimizing and understanding the non-convex optimization of these network architectures.

**Weaknesses:**

1. The major concern of the paper is that the technical contribution of the paper is limited. It is mostly built upon previous optimistic rate results ([Zhou et al. 2021, Zhou et al. 2022]). Upon reading the appendix and related literature, the reviewer is on the side that the paper serves as a simplification and rewritten of some past results (except the fact that there is a generalization of previous results by including an extra (network) parameter $\theta$). Still, the generalization results over general square-root Lipschitz loss is interesting, but need to note that there might not be enough technical contribution in the paper for the main theorem. The norm bounds in the applications are novel.

2. The paper itself is not presented in a way to highlight its contribution either. The paper has done a good job for the literature review, and it has spent more space introducing the results instead of providing intuitions for why the results hold true. This can be understandable as the space for the main submission is rather limited and there are many theorems to be stated in a single paper. Still, the reviewer thinks it can be beneficial in spending a bit more space on proof outlines.

3. There are some other limitations of the work, including the Gaussian feature assumption and the multi-index model for neural networks. Gaussian feature assumption can be served as a good starting assumption for the theory and the authors have addressed the limitation of this assumption appropriately. For neural networks, the multi-index model is a limited shallow neural network. In general, the Rademacher complexity for deep neural networks is normally only a loose estimation and thus results in unrealistic generalization bounds. Still, the multi-index model has been studied extensively recently, so the concern for the limitation here is just a minor one.

**Questions:**

1. How should I think about the technical contribution of the paper? Specifically, how should I consider the difference between the proof here for the square-root Lipschitz functions and the proof for a general Moreau envelope?

2. The generalization results over single-index model is interesting, where you have shown that $\max_{j \in [N]} |\sum_{i = 1}^j a_i| || w ||$ is a good complexity measure. Potentially combined with known results in the single index model, how large should this quantity be? In some way, I think this can be considered as future work to achieve end-to-end results for the weight-tied neural networks so should be fine if there is no clear answer here. Just trying to understand if the generalization bound we achieve here is tight in some sense.

**Limitations:**

The authors have addressed the limitation of the paper appropriately.

---

> ### Author Rebuttal · Authors · 2023-08-09
>
> Thanks for your feedback!
> > How should I think about the technical contribution of the paper? Specifically, how should I consider the difference between the proof here for the square-root Lipschitz functions and the proof for a general Moreau envelope?
>
> First, let us answer the specific question about the proof of the optimistic rates bound. Comparing the proof of the generalization bound with the Moreau envelope framework, we definitely needed a new analysis to avoid having unecessary assumptions involving the Moreau envelopes of f appear in Theorem 1. The new proof still uses key ingredients like the GMT, but after applying the GMT we have to be careful. Essentially, instead of trying to solve the auxiliary problem exactly, we show how to bound the auxiliary problem using analytic properties of square-root Lipschitz functions (in particular, allowing us to appeal to the useful calculus lemmas 8 and 9 in the appendix).
>
> Next, we would like to discuss the key technical contributions of our paper are, which do not end with establishing the optimistic rates bound. As a reminder, a very impressive theoretical understanding of benign overfitting in linear models (e.g. kernel machines) has emerged over the past few years. But studying (even mildly) nonlinear models has been a significant challenge since they do not seem as amenable to random matrix theory methods. One of the most important (and surprising!) realizations we had in this work was how to analyze benign overfitting in nonlinear models like phase retrieval & ReLU regression. At a technical level, this comes from the discovery of 1. the construction described in equation (8) which gives a low-norm interpolator for these nonlinear models, and 2. the closely related discovery of the correct ‘consistent losses’ for these problems. This was not at all obvious a priori! (E.g. we are not aware of anybody discovering the consistent loss (13) for interpolating ReLU regression before this work, or realizing that the Bartlett et al conditions for linear regression should also be sufficient for ReLU regression.) The extensions of the theory to matrix sensing, simple neural networks, and the non-gaussian setting of section 7 are also serious new contributions to the literature which cannot be obtained from previous work. In summary, the fact that sqrt-lipshitz losses satisfy a very sharp optimistic rates bound was one key finding, but the fact that combining this bound with the _right_ choice of sqrt-lipschitz losses lets us understand so many new phenomena is the deeper conceptual message of this work.
>
> > The generalization results over single-index model is interesting, where you have shown that max�∈[�]|∑�=1���|||�|| is a good complexity measure. Potentially combined with known results in the single index model, how large should this quantity be? In some way, I think this can be considered as future work to achieve end-to-end results for the weight-tied neural networks so should be fine if there is no clear answer here. Just trying to understand if the generalization bound we achieve here is tight in some sense.
>
> End-to-end results are definitely an interesting direction for future work, for example combining our generalization theory techniques with some of the algorithmic ideas in the literature. As far as tightness, if we choose the b_i appropriately then we can ensure that most of the data falls into a single linear region of the network, and \sum_{i = 1}^j a_i \|w\| for the corresponding value of j will be the norm of the corresponding linear predictor on this region. So in this case the bound reproduces the existing sharp norm-based generalization bound for linear models from previous work (which in turn was shown to recover benign overfitting in linear models etc.), and in this sense the bound seems pretty sharp.

---

> > ### Comment · Reviewer_8w1y · 2023-08-21
> > **Score Changed**
> >
> > The reviewer thanks the authors for their detailed explanation. I fully agree with the authors that all the applications discussed in the paper are interesting and important. As a result, even though I still consider the technical contribution of the sharp generalization bound with sqrt-Lipschotz losses to be limited (the connection and the result itself, are for sure very attractive), the paper provides a nice perspective in studying nonlinear models. I am willing to increase my score from 6 to 7.

---

### Official Review · Reviewer_949a · 2023-07-26

**Soundness:** 3 good
**Presentation:** 3 good
**Contribution:** 2 fair
**Rating:** 4
**Confidence:** 2

**Summary:**

In this paper, generic uniform convergence guarantees are provided for Gaussian data in terms of the Rademacher complexity of the hypothesis class and the Lipschitz constant of the square root of the scalar loss function. Square-Root Lipschitz Loss is an important class of loss function to study because it is a suitable loss to study interpolation learning in cases like phase retrieval and matrix sensing. The authors obtain an optimistic rate in theorem 1 of the paper and the analysis is mostly inspired by the previous work of  Zhou et al. 2022. Then the authors discuss multiple use cases and provide generalization error bound for over-parametrized phase retrieval, matrix sensing, and single index weight tied neural network.

**Strengths:**

The paper is technically a good paper. Though I am not an expert in this area but it looks to me like, this paper modified the analysis in the previous recent work by Zhou et al. 2022 to obtain new overfitting results.

**Weaknesses:**

I have a few questions and some might seem very trivial and please correct me if I am wrong.
1. I can see that the paper utilizes the Gaussian minimax theorem that is explained in Appendix B.2. However, this still seems like a very restricted setting to me. What happens when one assumes that data is coming from a more general distribution?  Previous results (though not applicable for square roor Lipschitz loss) have no major distributional assumption.
2. I am also curious to know how realistic is overparametrization setting in phase retrieval.
3. I am also curious to understand the results of the single index neural network model as the paper cited in this work Biettie et al and other related papers in the domain seem to have the stronger guarantee of recovering the direction using spherical SGD or gf. In this paper, the guarantees are for generalization errors. Is there a way to compare these two results as the results in the other papers where there recover the optimal direction sounds stronger to me?

As a minor comment, it would also be great if the authors can discuss and compare the results in section 5 and 6 with previously known results. A discussion would be very helpful.





**Questions:**

Please see above.

**Limitations:**

Not applicable.

---

> ### Author Rebuttal · Authors · 2023-08-09
>
>
> Thanks for the feedback.
>
> The Gaussianity assumption is indeed a significant restriction, and deviates from much of the classical distribution-free statistical learning literature.  But assuming Gaussianity allows obtaining much tighter guarantees, with tight numerical constants (see [Koehler et al ‘21, Zhou et al ‘21]).  Gaussianity is also widely assumed and used in analysis of many statistical learning and inference problems, e.g. sparse recovery [e.g. Stojnic ‘13, Chandrasekaran et al ‘12 https://arxiv.org/abs/1012.0621, …], phase retrieval [e.g. Mondelli and Montanari ‘18, Barbier et al ‘19], logistic regression [e.g. Candes-Sur ‘20 https://arxiv.org/abs/1804.09753], many other works such as those using Approximate Message Passing, and even analysis of deep learning [e.g. Soltanolkotabi ‘17 https://arxiv.org/abs/1705.04591] (although most of these were also studied, usually with weaker guarantees, without assuming Gaussianity).  Furthermore, much of the statistical physics based analysis relies on “Gaussian universality”, which essentially also relies on the data behaving as if it was Gaussian [e.g. see Hu-Lu ‘23, Bayati et al ‘12 https://arxiv.org/pdf/1207.7321.pdf for some discussion and related rigorous results].  And so, although we agree this is a significant restriction, and it would be very interesting to relax this assumption, we still believe that results relying on Gaussianity are interesting and useful, both on their own right, and as a step toward more general analysis.
>
> Specifically in the context of this work, by working with Gaussian data we were able to discover several interesting phenomena (e.g. benign overfitting in ReLU regression) which were not at all obvious beforehand (for example, that the consistent loss for interpolation in ReLU regression is (13), that this loss is sqrt-lipschitz, and that it satisfies an “optimistic rate” bound). With a view towards understanding all of these new phenomena beyond the Gausian setting, we have included Section 7, which reveals a situation where a naive generalization of our optimistic rates bound to non-gaussian data is false, but a correct & more sophisticated generalization (equation (28)) works.
>
> > I am also curious to know how realistic is overparameterization setting in phase retrieval.
>
> The main significance of the phase retrieval model is that it is one of the simplest and most canonical nonlinear models, and so it is valuable to rigorously understand benign overfitting there before proceeding to more complex models like deep networks. Previous works have certainly studied the behavior of phase retrieval in high-dimensional limits with many parameters where overfitting occurs (e.g. cited work of Maillard et al) and understanding the behavior under overparameterization seems like a natural goal just as it was with linear regression.
>
> > I am also curious to understand the results of the single index neural network model as the paper cited in this work Biettie et al and other related papers in the domain seem to have the stronger guarantee of recovering the direction using spherical SGD or gf. In this paper, the guarantees are for generalization errors. Is there a way to compare these two results as the results in the other papers where there recover the optimal direction sounds stronger to me?
>
> Besides that we are both studying weight-tied neural networks, the results seem to be incomparable. Ours is a generalization bound, so it would apply to any algorithm/estimator and we expect it is pretty quantitatively sharp (see also the response to reviewer 8w1y). On the other hand, we didn’t analyze any polynomial time algorithm. The previous works you mentioned analyzed a particular algorithm and proved some guarantees for it (their main goal is algorithmic efficiency, so they did not carefully analyze features like constant factors in the number of samples which we care about in this work). There is probably a lot of interesting future work that could be done combining these types of algorithmic + statistical analyses.
>
> > As a minor comment, it would also be great if the authors can discuss and compare the results in section 5 and 6 with previously known results. A discussion would be very helpful.
>
> As far as section 5 is concerned, there was a lot of previous work on matrix sensing, ReLU regression etc. but we are the first to establish these types of results, i.e. to prove (1) sufficient conditions for benign overfitting in these models and (2) as the key ingredient to achieve 1, prove very sharp generalization and norm bounds in these models. If we only wanted to prove _some_ generalization bound for say phase retrieval, we could apply standard tools from statistical learning theory like symmetrization+contraction, but they are very wasteful since our loss is not Lipschitz, but instead sqrt-Lipschitz.
>
> Regarding section 6, we have not seen a generalization bound of quite an analogous form (we are taking advantage of the weight-tied structure here). For non-weight-tied neural networks, it is possible to state generalization bounds in terms of the l1 norm of the weights (which could be bigger than the maximum partial sum which appears in our bound), and this type of analysis of neural networks dates at least back to the [Bartlett ‘96] reference.

---

### Decision · Program_Chairs · 2023-09-21

**Decision:**

Accept (poster)

**Comment:**

This is a nice paper which, to my taste, can be best summarized somewhat technically as trying to find the limits of the Gaussian minimax technique to interpolation / benign overfitting from prior works, and resulting in a rather simple presentation which applies to many settings and moreover leads to some interesting further discussion and insight, for instance the counterexample in section 7. I am happy to recommend acceptance, and recommend the authors use the extra camera ready page to reflect all reviewer discussion.